# Proteomic and clinical biomarkers for acute mountain sickness in a longitudinal cohort

Jing Yang [1,2,3,4,8], Zhilong Jia [3,4,5,8 ✉], Xinyu Song [3,5], Jinlong Shi[2,3], Xiaoreng Wang[6], Xiaojing Zhao[4,7] & Kunlun He [1,2,3,4 ✉]

Ascending to high-altitude by non-high-altitude natives is a well-suited model for studying acclimatization to extreme environments. Acute mountain sickness (AMS) is frequently experienced by visitors. The diagnosis of AMS mainly depends on a self-questionnaire, revealing the need for reliable biomarkers for AMS. Here, we profiled 22 AMS symptom phenotypes, 65 clinical indexes, and plasma proteomic profiles of AMS via a combination of proximity extension assay and multiple reaction monitoring of a longitudinal cohort of 53 individuals. We quantified 1069 proteins and validated 102 proteins. Via differential analysis, machine learning, and functional association analyses. We found and validated that RET played an important role in the pathogenesis of AMS. With high-accuracies (AUCs > 0.9) of XGBoost-based models, we prioritized ADAM15, PHGDH, and TRAF2 as protective, predictive, and diagnostic biomarkers, respectively. Our findings shed light on the precision medicine for AMS and the understanding of acclimatization to high-altitude environments.

[1] Medical School of Chinese PLA, Chinese PLA General Hospital, Beijing, China. [2] Research Center for Medical Big Data, Medical Innovation Research Division of Chinese PLA General Hospital, Beijing, China. [3] Key Laboratory of Biomedical Engineering and Translational Medicine, Ministry of Industry and Information Technology, Chinese PLA General Hospital, Beijing, China. [4] Beijing Key Laboratory of Chronic Heart Failure Precision Medicine, Chinese PLA General Hospital, Beijing, China. [5] Center for Artificial Intelligence in Medicine, Medical Innovation Research Division of Chinese PLA General Hospital, Beijing, China. [6] Laboratory of Radiation Injury Treatment, Medical Innovation Research Division, PLA General Hospital, Beijing, China. [7] Translational Medicine Research Center, Medical Innovation Research Division of Chinese PLA General Hospital, Beijing, China. [8]These authors contributed equally: Jing Yang, Zhilong Jia. ✉email: jiazhilong@plagh.org; kunlunhe@plagh.org

Acclimatization and environmental adaptation play important roles in human activity and survival. High-altitude is a typical extreme environment, providing a well-suited model for studying acclimatization. Millions of people travel or operate in high-altitude areas every year. People traveling to high-altitude regions have an increased risk of developing high-altitude illness, such as acute mountain sickness (AMS)[1], high-altitude pulmonary edema (HAPE)[2], and chronic mountain sickness (CMS)[3]. The capacity for acclimatization to the high-altitude varies for different individuals.

AMS, one of the most common high-altitude illnesses, occurs when individuals living at low altitudes ascend to high altitudes (above 2500 m)[2]. The diagnosis of AMS depends on questionnaire-based diagnostic instruments, among which the Lake Louise Acute Mountain Sickness Score (LLS) is widely used[4]. The typical symptoms of AMS are headache, gastrointestinal symptoms (poor appetite, nausea, and/or vomiting), dizziness, and fatigue. Without effective treatment, AMS may progress to life-threatening high-altitude cerebral edema or HAPE[2]. The incidence of AMS varies from 25 to 94%, affected by multi-factors, such as anxiety[5,6], altitude attained[2], rate of ascent[7], individual susceptibility[8], and preacclimatization[9]. Moreover, the lack of reliable diagnostic criteria makes AMS complicated.

The pathogenesis of AMS remains unclear and is under debate. Several hypotheses have been proposed for the pathophysiological mechanism of AMS, such as the induction of vasogenic and/or intracellular cerebral edema[10–12], increases in vascular permeability due to higher levels of oxidative stress[13] and inflammation[11], increases in vascular endothelial growth factor levels induced by hypoxia-inducible transcription factors[14], and metabolic demand[15,16]. Because these pathophysiological mechanisms cannot completely characterize AMS, it is difficult to identify biomarkers according to these methods.

The proteome is suitable for exploring the pathogenesis and discovering biomarkers of AMS. Previous studies on the proteome of AMS mainly used enzyme-linked immunosorbent assay, gel electrophoresis and mass spectrometry (MS) technology[17–22]. Notably, the gel electrophoresis-based method has several drawbacks, such as quantitative reproducibility and a biased proteome profile[23]. Julian et al.[18] found that the abundance of antioxidant proteins was higher in patients with AMS but not in individuals with resistance to AMS using a 20-volunteer cohort. By employing isobaric tags for relative and absolute quantitation MS, Lu et al.[15] revealed that proteins related to the tricarboxylic acid cycle (for example, pyruvate dehydrogenase E1 subunit alpha 1 (PDHA1), succinate dehydrogenase complex flavoprotein subunit A (SDHA), and succinate-CoA ligase GDP/ADP-forming subunit alpha (SUCLG1)), glycolysis (such as fructose-bisphosphatase 1 (FBP1), aldolase, fructose-bisphosphate A (ALDOA), and phosphoglycerate kinase 1 (PGK1)), the ribosome, and the proteasome (for example, proteasome 26S subunit, ATPase 3 (PSMC3)) were significantly suppressed in the AMS-resistant group compared with their levels in the AMS-susceptible group. The levels of anti-inflammatory and/or anti-permeability factors, such as interleukin 1 receptor antagonist (IL-1RA), heat shock protein-70 (HSP-70), and adrenomedullin, are higher in AMS-resistant subjects than in AMS-susceptible subjects, whereas the levels of the chemotactic factors C-C motif chemokine ligand 2 (CCL2) and tumor necrosis factor alpha (TNF-α) are independent of the AMS status[11]. Kevin et al.[24] found that Angiopoietin-like 4 (ANGPTL4) and resistin were increased at a high altitude in AMS patients. However, the findings obtained in most of these studies were not validated. Padhy et al.[20] observed a lower abundance of angiotensinogen and angiotensin II in high-altitude natives by matrix-assisted laser desorption/ionization-time of flight/time of flight (MALDI_TOF/TOF). Moreover, studying the proteome of high-altitude natives could also shed light on AMS in non-high-altitude natives. Du et al.[25] performed a tandem mass tag-label-based (TMT-label-based) plasma proteomic analysis and found that C-C motif chemokine ligand 2 (CCL18), complement C9 (C9), and S100 calcium binding protein A9 (S100A9) were upregulated and that histidine rich glycoprotein (HRG) and coagulation factor XI (F11) were downregulated in high-altitude natives compared with non-high-altitude natives at high altitude. Accordingly, a systematic study of the pathogenesis, candidate therapeutic targets, and protective, predictive and diagnostic biomarkers of AMS using a larger cohort and new advanced proteome technology is needed.

Proximity extension assay (PEA) technology uses antigen-antibody binding and quantitative real-time polymerase chain reaction (qPCR) technology to perform qualitative and quantitative analysis of proteins and has been widely used for the study of several diseases, such as COVID-19[26,27] and cardiovascular disease[28,29]. The Olink panels cover biomarkers of crucial diseases and proteins of important biological processes involving different systems, allowing us to profile thousands of proteins using only a small volume of plasma. Moreover, multiple reaction monitoring (MRM) is an MS-based method targeting selective peptides for protein detection and quantitation with good reproducibility and sensitivity. MRM technology is suitable for the validation of candidate biomarkers. The combination of antibody-based PEA and MS-based MRM technologies will largely eliminate false-negative signals. Therefore, we applied both PEA and MRM technologies in our study to obtain reliable and robust results.

In this study, we systematically explored AMS using the proteomes, 65 clinical indexes, and 20 AMS symptom phenotypes of 106 plasma samples from a Chinese Han cohort consisting of 53 participants (Fig. 1a). We characterized the protein profile of 10 participants with AMS at low and high altitudes using Olink's PEA technology (Fig. 1b) and validated 102 key proteins using MRM proteomics technology with expanded samples and groups (Fig. 1c). Moreover, we identified candidate pathogenesis-related proteins and protective, predictive, and diagnostic biomarkers of AMS via statistical analysis and a machine learning-based model (Fig. 1d). In addition, we associated these proteins with pathways, AMS symptom phenotypes, and clinical indexes to dissect the function of these proteins in AMS with the aim of redefining AMS with proteins and clinical indexes (Fig. 1e). Our study illuminates potentially important pathogenesis-related proteins, robust therapeutic targets, and predictive and diagnostic biomarkers of AMS with the aim of promoting an improved understanding and redefinition of AMS and high-altitude acclimatization.

## Results

**Study design for exploring AMS**. We recruited a cohort of 53 individuals and collected 106 plasma samples (53 pairs) from the participants living at low and at high altitude for 1–4 days. All the participants completed a questionnaire within 12–24 h of arrival at high altitude, which included 22 AMS symptom phenotypes (Supplementary Data 1), when they were at high altitude prior to blood collection. Among the tested symptom phenotypes, headache and 5 other symptom phenotypes were used to calculate the Louise Lake Acute Mountain Sickness Score (see Methods). Among the 53 subjects, 30 subjects (57%) were diagnosed as AMS. To perform comprehensive proteomic profiling of AMS, we identified and quantified 1069 proteins in 20 plasma samples of AMS subjects using PEA technology at the discovery stage and 102 proteins in 106 plasma samples using MRM technology at the validation stage. The discovery stage included two groups from a cohort with ten individuals with AMS: individuals with AMS at high-altitude (AMS4k) and the same individuals at low-altitude (AMS1k). The validation stage

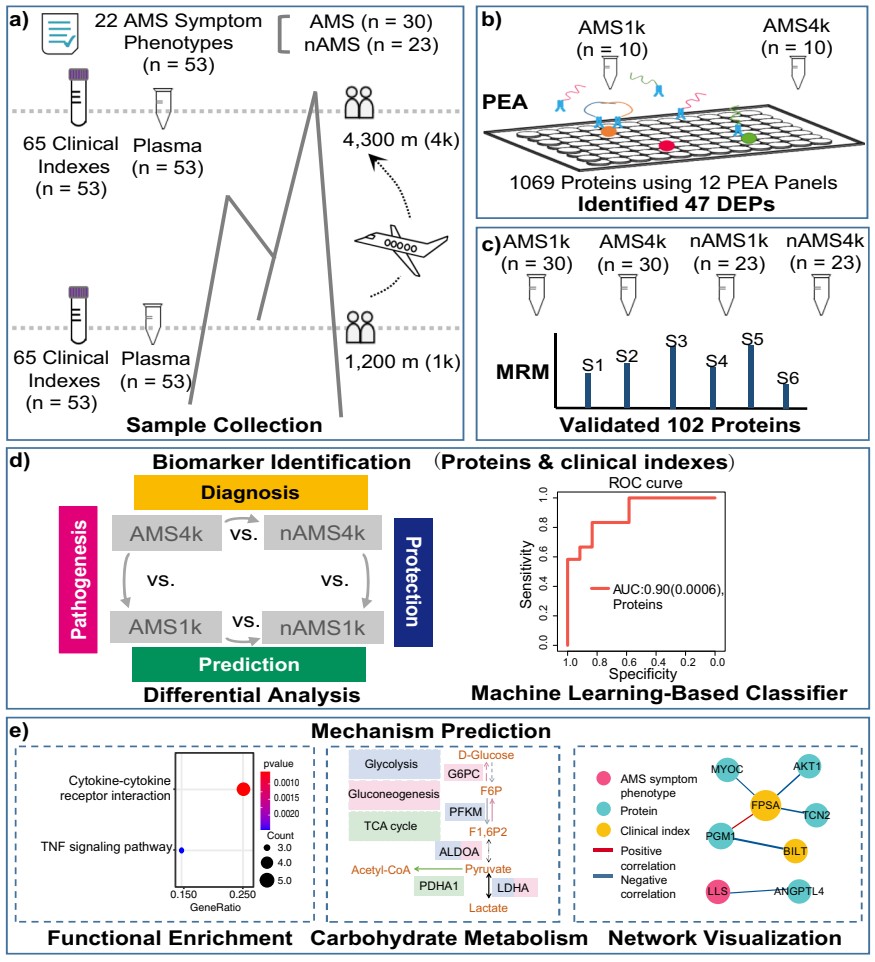

**Fig. 1 Study design. a** Sample collection. We recruited 53 subjects, collected plasma samples and recorded clinical indexes at low altitudes (1 km) and high altitudes (4 km), and recorded 22 AMS symptom phenotypes at high altitude. According to the Louise Lake Score, the participants were separated into 30 individuals with AMS (AMS) and 23 individuals without AMS (nAMS). **b** At the discovery stage, proximity extension assay (PEA) technology identified 47 plasma differentially expressed proteins (DEPs) between the ten AMS4k and ten AMS1k samples. **c** At the validation stage, multiple reaction monitoring (MRM) technology validated 102 proteins in four groups of samples (AMS1k: $n = 30$, AMS4k: $n = 30$, nAMS1k: $n = 23$, nAMS4k: $n = 23$). **d** Differential analysis and machine learning-based classifiers with proteins and/or clinical indexes were used in the pathogenesis (pink background, comparison between the AMS4k and AMS1k groups), protection (dark blue background, comparison between the nAMS4k and AMS1k groups), prediction (dark green background, comparison between the AMS1k and nAMS1k groups), and diagnosis (yellow background, comparison between the AMS4k and nAMS4k groups) comparisons. **e** Functional enrichment analysis and network analysis were performed to illustrate the mechanism.

included four groups from a cohort with 30 individuals with AMS and 23 individuals without AMS: individuals with AMS at high-altitude (AMS4k), the same individuals at low-altitude (AMS1k), individuals without AMS at high-altitude (nAMS4k), and the same individuals at low-altitude (nAMS1k). We defined "pathogenesis" as a comparison between individuals with AMS at high-altitude (4k) and low-altitude (1k) in our nature-interventional study. The proteomic signature observed in individuals with AMS at 4k relative to the same individuals at 1k could be used to describe the pathogenesis of the AMS disease. We defined "protection" as a comparison between individuals without AMS at 4k and 1k in our nature-interventional study. The proteomic signature observed in individuals without AMS at 4k relative to the same individuals at 1k could be used to describe protection from AMS disease. We defined "prediction" as a comparison between individuals with AMS at 1k and individuals without AMS at 1k in our nature-interventional study. The proteomic signature observed in individuals with AMS vs. individuals without AMS at 1k (before the intervention) could be used to describe the prediction of AMS disease. We defined "diagnosis" as a comparison between individuals with AMS at 4k and individuals without AMS at 4k in our nature-interventional study.

The proteomic signature observed in individuals with AMS vs. individuals without AMS at 4k (once the change in altitude had already been conducted) could be used to describe the diagnosis of AMS disease.

**PEA-based identification of proteins involved in AMS at the discovery stage.** At the discovery stage, we measured 21,600 measurements, consisting of 1069 proteins per subject per condition in ten subjects (10 AMS) and two conditions (1k and 4k), and 22 AMS phenotypes in ten subjects and one condition (4k), to explore the pathogenesis and therapeutic targets of AMS using 12 PEA panels (Supplementary Data 2). The average age and LLS score at high altitude were 18.8 and 5.2, respectively. Strict quality control identified 887 proteins (see Methods). Based on a $q$ value < 0.05, we identified 47 differentially expressed proteins (DEPs), which included 40 upregulated and 7 downregulated proteins (Fig. 2a and Supplementary Data 3). ANGPTL4, matrix metallopeptidase 3 (MMP3), and fibroblast growth factor 23 (FGF23) were upregulated in AMS4k, whereas carbonic anhydrase 1 (CA1), carbonic anhydrase 2 (CA2), and the chemokine CCL2 were downregulated (Fig. 2a).

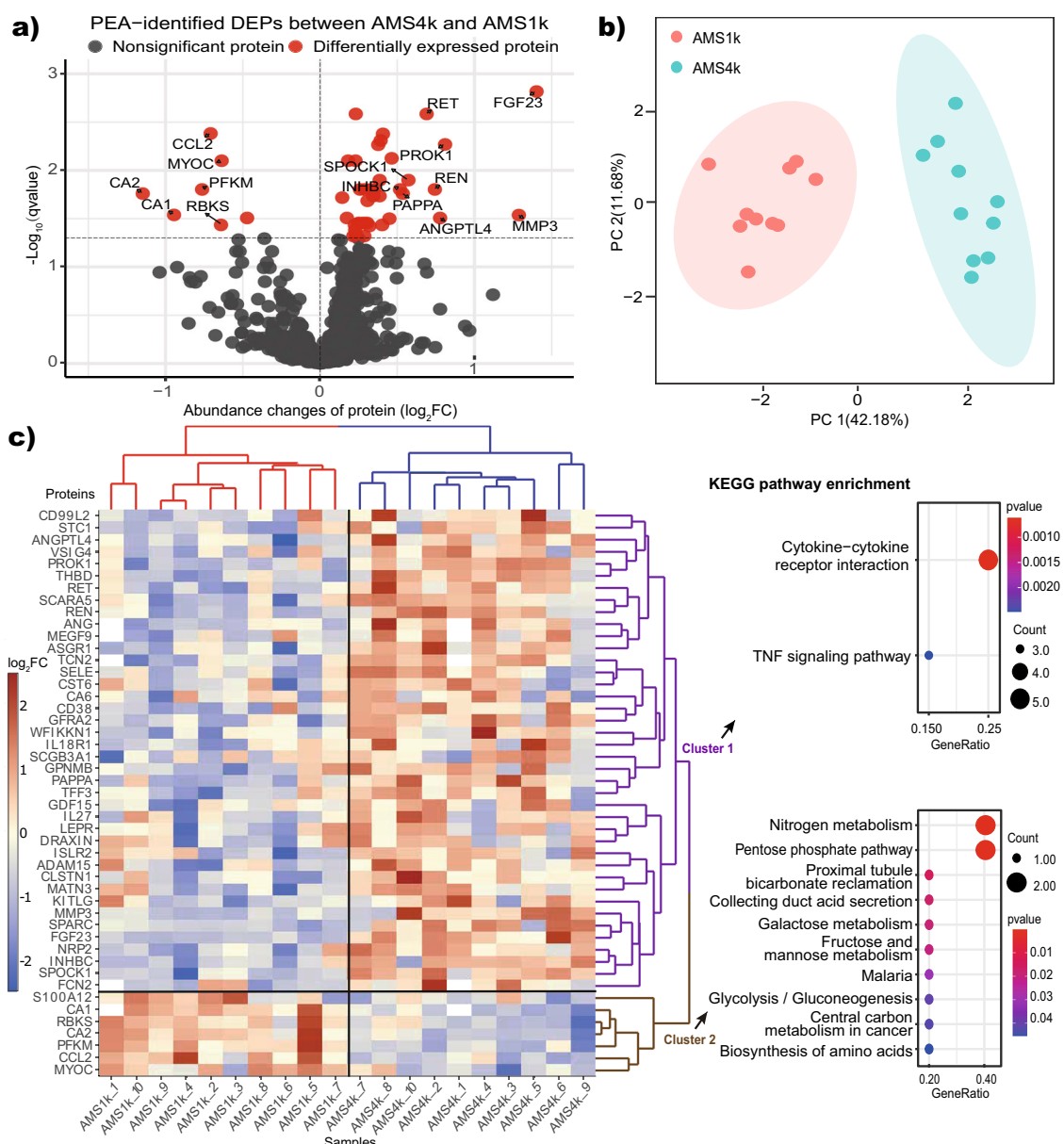

**Fig. 2 PEA-identified proteins and biological functions involved in AMS. a** Volcano plot of PEA-identified DEPs between the AMS4k ($n = 10$) and AMS1k ($n = 10$) groups. Proteins with no statistical significance ($q$ value $\geq 0.05$, gray dots) and DEPs ($q$ value $< 0.05$, red dots) with absolute value of $\log_2$FC larger than 0.5 are labeled with gene symbols. **b** PCA plot based on PEA-identified DEPs. The AMS1k (light red dots) and AMS4k (sky-blue dots) samples were well separated by PCA using the PEA-identified DEPs without missing values ($n = 42$). **c** Abundance heatmap and KEGG pathway enrichment diagram of the PEA-identified DEPs. The scaled abundance heatmap of DEPs between the AMS4k and AMS1k groups per sample is shown on the left side. The hierarchical cluster analysis in the heatmap showed that the 47 DEPs could clearly distinguish the AMS4k group (red hierarchical tree) and AMS1k group (blue hierarchical tree), and the DEPs could be separated into cluster 1 (purple hierarchical tree) and cluster 2 (brown hierarchical tree). Dot plots of the KEGG enrichment of the two clusters with $p$ values $< 0.05$ are presented on the right side. Cytokine–cytokine receptor interactions and the TNF signaling pathway were enriched in cluster 1 (purple fonts), and nitrogen metabolism and the pentose phosphate pathway were mainly enriched in cluster 2 (brown fonts).

The DEPs were sufficiently powered to clearly distinguish the AMS1k and AMS4k groups. Principal component analysis (PCA) showed that these proteins clearly distinguished the AMS4k group from the AMS1k group, and the first principal component accounted for 42.18% of the variance (Fig. 2b). A clustering analysis also showed a clear separation between the two groups (Fig. 2c). Accordingly, we considered the 47 DEPs to be a primary clue for exploring the therapeutic targets of and protective, predictive, and diagnostic biomarkers for AMS.

**Identification of biological functions involved in AMS at the discovery stage.** By performing a functional enrichment analysis of the upregulated and downregulated proteins, we identified pathways and Gene Ontology (GO) terms related to AMS. Energy metabolism pathways, such as nitrogen metabolism, the pentose phosphate pathway, and glycolysis/gluconeogenesis, were significantly downregulated in cluster 2, whereas cytokine–cytokine receptor interaction and TNF signaling pathways were significantly upregulated in cluster 1 (Fig. 2c, Supplementary Data 4).

**Table 1 Baseline characteristics of the 53 subjects.**

|  | nAMS (n = 23) | AMS (n = 30) | p value |
|---|---|---|---|
| Sex |  |  |  |
| Male | 23 (100%) | 30 (100%) | NA |
| Female | 0 (0%) | 0 (0%) |  |
| Age (years) |  |  |  |
| Mean (SD) | 19.0 (1.30) | 19.3 (1.23) | 0.528 |
| Median [min, max] | 19.0 [17.0, 22.0] | 19.0 [17.0, 22.0] |  |
| Height (cm) |  |  |  |
| Mean (SD) | 171 (4.70) | 173 (5.47) | 0.157 |
| Median [min, max] | 170 [162, 180] | 173 [165, 185] |  |
| Weight (kg) |  |  |  |
| Mean (SD) | 63.1 (5.05) | 65.9 (7.00) | 0.0996 |
| Median [min, max] | 62.0 [54.0, 72.0] | 66.0 [53.0, 84.0] |  |
| LLS |  |  |  |
| Mean (SD) | 0.696 (0.703) | 4.3 (1.49) | <0.001 |
| Median [min, max] | 1 [0, 2] | 4 [3, 9] |  |

The GO enrichment analysis revealed that cell growth and response to stimuli, such as hypoxia and cAMP, were upregulated in cluster 1 (Supplementary Fig. 1), whereas bicarbonate transport, monocyte chemotaxis, and monosaccharide catabolic processes were downregulated in cluster 2 (Supplementary Fig. 1).

Carbonic anhydrase (CA) family proteins, including CA1, CA2, and carbonic anhydrase 6 (CA6), are actively involved in AMS via the carbonate dehydratase process of the nitrogen metabolism pathway. In total, three types of cytokines, namely, the IL6 receptor family (LEPR), IL-1 receptor family (IL18R1), and TGF-β family (growth differentiation factor 15 (GDF15) and inhibin beta C subunit (INHBC)), which are involved in the cytokine–cytokine receptor interaction pathway, were upregulated. The results also revealed the upregulation of IL18R1, selectin E (SELE), and MMP3, which are involved in the TNF signaling pathway. These findings indicate that immunological and inflammatory responses and energy metabolism are actively involved in the pathogenesis of AMS.

**Validation via multiple reaction monitoring**. To confirm the association between PEA-identified DEPs and AMS, we validated 102 proteins in 106 samples and 4 groups (AMS1k, AMS4k, nAMS1k, and nAMS4k) using an MRM MS-based proteome platform. No significant differences in age, height, or weight were found between the individuals with and without AMS, whereas the LLS scores at high altitude were significantly lower in the individuals without AMS than in the individuals with AMS ($0.696 \pm 0.703$ vs. $4.3 \pm 1.49$, $p$ value < 0.001) (Table 1). We extended the 47 proteins to 102 proteins by adding 55 proteins involved in several pathways, such as the pentose phosphate and glycolysis pathways. Finally, we profiled 102 proteins (538 fragment ions) in 53 paired plasma samples (30 AMS4k, 30 AMS1k, 23 nAMS4k, and 23 nAMS1k) from 53 male Chinese Han subjects using MRM technology (Supplementary Fig. 2). The comparisons between the AMS4k and AMS1k, nAMS4k and nAMS1k, AMS1k and nAMS1k, and AMS4k and nAMS4k groups could be used to illustrate the pathogenesis, protection, prediction, and diagnosis of AMS, respectively (Fig. 1d).

**Key proteins involved in the pathogenesis of AMS**. The comparison between the AMS4k and AMS1k groups could be used to identify AMS pathogenesis-related proteins. We confirmed that 23 out of 47 proteins (49%) exhibited changing trends that were consistent with those found for the PEA-identified DEPs (Supplementary Data 3, Supplementary Fig. 3). Among these proteins, matrilin 3 (MATN3), myocilin (MYOC), ret proto-oncogene

(RET), and S100 calcium binding protein A12 (S100A12) were significantly different (Fig. 3a, Supplementary Fig. 4a, b), which validates their potential involvement in the response to AMS. Myocilin, which is encoded by MYOC, is involved in cytoskeletal function[30]. MATN3 promotes the expression of HIF-1α[31], the main factor in HIF-1α pathway, in response to hypoxia and inflammation[32,33]. S100A12 is a calcium-, zinc- and copper-binding protein that plays a prominent role in the regulation of inflammatory processes and the immune response[34]. RET, a transmembrane receptor and member of the tyrosine-protein kinase family of proteins, plays a role in cell differentiation, growth, migration, and survival. Notably, RET showed an opposite changing trend between nAMS4k and nAMS1k, although this difference was not statistically significant (Fig. 3a). In addition, the HIF-1 signaling pathway, glycolysis/gluconeogenesis, and carbon metabolism were enriched in DEPs identified by MRM (Supplementary Fig. 5a and Supplementary Data 4). The HIF-1 signaling pathway is associated with AMS[35].

**Protective proteins for AMS**. The comparison between the nAMS4k and nAMS1k groups could be used to explore protective proteins and could shed light on the pathophysiological mechanism and prevention of AMS. We identified 29 protective DEPs between the nAMS4k and nAMS1k groups (Supplementary Data 3). Overall, the trends obtained for ADAM metallopeptidase domain 15 (ADAM15), CD38 molecule (CD38), cystatin E/M (CST6), KIT ligand (KITLG), and thrombomodulin (THBD) between these two groups were the opposite of those obtained for these proteins between the AMS4k and AMS1k groups, as revealed by both PEA and MRM (Fig. 3a, Supplementary Fig. 4c, d), which indicates their potential protective role in preventing AMS and rapid acclimation to high altitude. Among these proteins, ADAM15 exhibited the highest degree of downregulation. In addition, the downregulated proteins CD38 and KITLG are involved in the hematopoietic cell lineage (Supplementary Fig. 5b and Supplementary Data 4), and KITLG is induced by HIF-1α under hypoxia in cancer cells[36] and plays a role in hematopoiesis and cell migration.

**Predictive biomarkers of AMS**. Predicting the occurrence of AMS at high altitudes is difficult but has great value for individuals living at low altitude. The comparison between the AMS1k and nAMS1k groups could identify predictive biomarkers. We identified 23 DEPs between the AMS1k and nAMS1k groups (Supplementary Data 3), and 17 of these DEPs showed the same trend as that obtained between the AMS4k and nAMS4k groups, showing the robustness of these proteins (Fig. 3a, Supplementary Fig. 6a). Filtering based on an absolute value of $\log_2$FC larger than 0.5 resulted in 6 upregulated proteins: phosphoenolpyruvate carboxykinase 1 (PCK1), phosphoglycerate dehydrogenase (PHGDH), ribokinase (RBKS), S100A12, solute carrier family 4 member 1 (SLC4A1), and secreted protein acidic and cysteine rich (SPARC). Among these DEPs, PCK1[37], and RBKS[38,39] are involved in the metabolism of carbohydrates. These results indicate that these proteins are candidate predictive biomarkers of AMS.

**Diagnostic biomarkers of AMS**. The comparison between the AMS4k and nAMS4k groups could be used to discover diagnostic biomarkers of AMS. We identified 28 DEPs between the AMS4k and nAMS4k groups (Supplementary Data 3). After filtering based on an absolute value of $\log_2$FC larger than 0.5, 10 upregulated proteins (ADP ribosylation factor 6 (ARF6), Epstein-Barr virus induced 3 (EBI3), GC vitamin D binding protein (GC), immunoglobulin superfamily containing leucine rich repeat 2 (ISLR2), MYOC, neuropilin 2 (NRP2), RBKS, RET, TNF

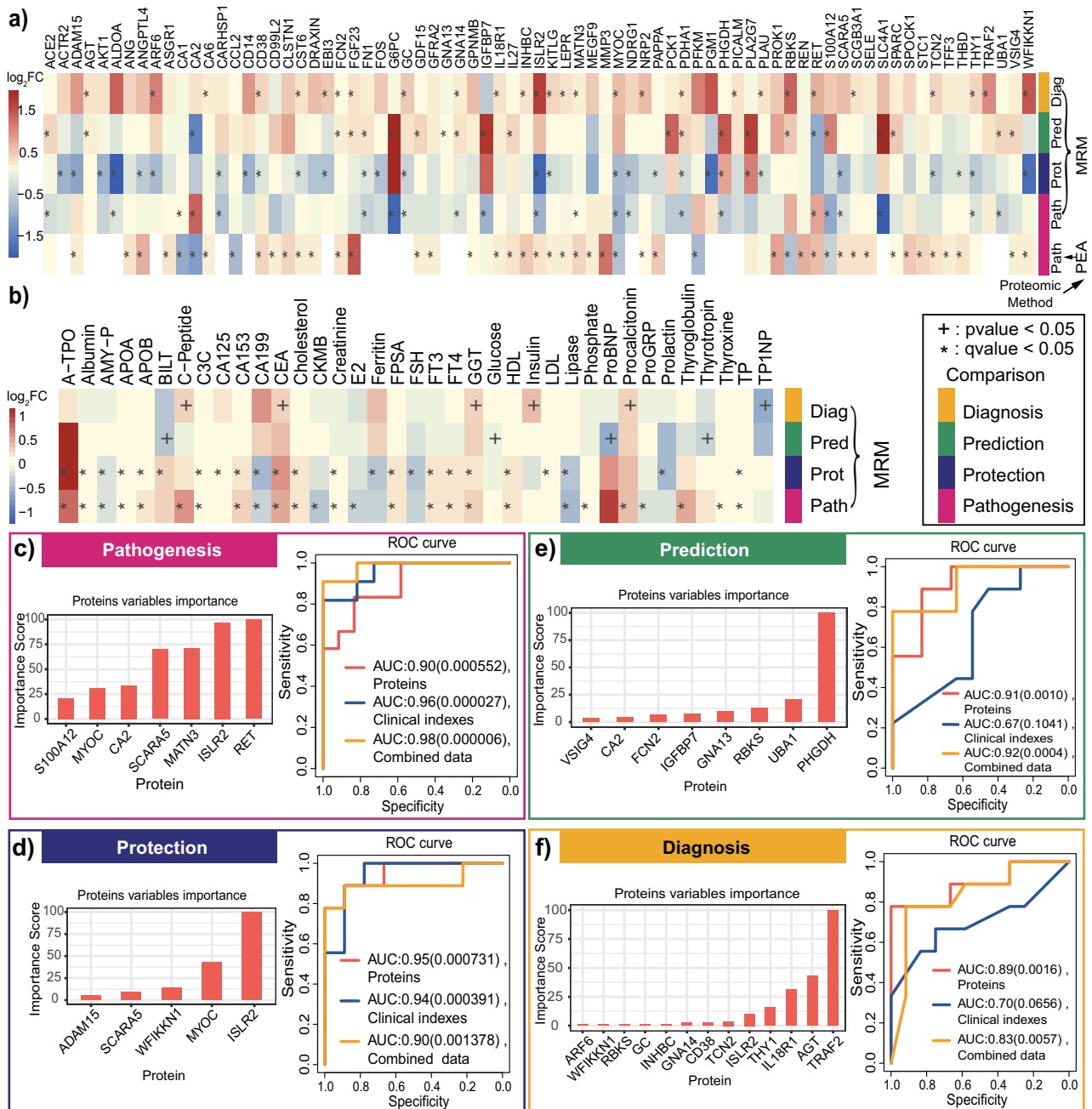

**Fig. 3 Differential analysis and classification with MRM-identified DEPs and clinical indexes.** Differential analysis and machine learning were performed with the identified proteins and clinical indexes in four comparison groups (pathogenesis: AMS4k ($n = 30$) and AMS1k ($n = 30$); protection: nAMS4k ($n = 23$) and nAMS1k ($n = 23$); prediction: AMS1k ($n = 30$) and nAMS1k ($n = 23$); and diagnosis: AMS4k ($n = 30$) and nAMS4k ($n = 23$). **a** Heatmap of the log$_2$FC of 75 PEA- or MRM-identified DEPs in the 5 comparison groups based on PEA or MRM. **b** Heatmap of the log$_2$FC of 37 differential clinical indexes involved in the 4 comparison groups. The heatmap depicts the regulation trends of each protein or clinical index with the legend at the left of the figure. The statistical methods were used paired/unpaired t test or Welch's t test. The differentially expressed proteins and clinical indexes are labeled with * (q value < 0.05) and +(p value < 0.05). **c** ROCs of the classification models with proteins (orange curve), clinical indexes (blue curve), or combined data (yellow curve) and bar plot of important proteins (orange bar) between AMS4k and AMS1k (pathogenesis, pink box). All the pathogenesis models with three types of features exhibited high accuracy with statistical significance, as demonstrated by the AUCs and p values. **d** ROCs of the classification models with proteins (orange curve), clinical indexes (blue curve), or combined data (yellow curve) and bar plot of important proteins (orange bar) between the nAMS4k and nAMS1k groups (protection, dark blue box). All protection models with three types of features show high accuracy with statistical significance. **e** ROCs of the classification models with proteins (orange curve), clinical indexes (blue curve), or combined data (yellow curve) and bar plot of important proteins (orange bar) between the AMS1k and nAMS1k groups (prediction, dark green box). The prediction models with proteins and combined data show high accuracy. **f** ROCs of the classification models with proteins (orange curve), clinical indexes (blue curve), or combined data (yellow curve) and bar plot of important proteins (orange bar) between the AMS4k and nAMS4k groups (diagnosis, yellow box). The diagnosis models with proteins and combined data exhibit high accuracy with statistical significance. The p value (in parentheses) of each AUC is shown after the AUC value. ROC receiver operating characteristic curve, AUC area under the curve.

receptor-associated factor 2 (TRAF2), and WAP, follistatin/kazal, immunoglobulin, kunitz and netrin domain containing 1 (WFIKKN1)) remained (Fig. 3a, Supplementary Data 3, Supplementary Fig. 6b). Among these proteins, EBI3, RET, and WFIKKN1 showed opposite regulation trends compared with those obtained from the comparison between the AMS1k and nAMS1k groups (Fig. 3a), which shows their higher likelihood of being diagnostic biomarkers. Notably, RET exhibited the same significant regulation trends between the AMS4k and AMS1k groups, indicating its active role in the pathogenesis of AMS. ISLR2 is involved in axon guidance in brain development, and ISLR2 deficiency leads to severe hydrocephalus in mice[40]. ISLR2 also shows a genetic interaction with RET[41].

**High AMS classification accuracy**. Four tree-based XGBoost machine-learning models showed good performance in distinguishing AMS4k from AMS1k, nAMS4k from nAMS1k, AMS4k from nAMS4k, and AMS1k from nAMS1k. We used 10-fold cross-validation and the DEPs measured by MRM assays to enhance the robustness and performance of the models. The area under the curve (AUC) of the pathogenesis model with a panel of 7 proteins (RET, ISLR2, MATN3, scavenger receptor class A member 5 (SCARA5), CA2, MYOC, and S100A12) to distinguish AMS1k and AMS4k was 0.90 ($p$ value < 0.001, Fig. 3c). Notably, RET, MYOC, MATN3, and S100A12 were also candidate pathogenesis-related proteins according to differential abundance analysis (Fig. 3a). MYOC, a type of secreted glycoprotein that participates in cell adhesion, cell-matrix adhesion, cytoskeleton organization, and cell migration, was reported to be reduced after incubation under hypoxia in trabecular meshwork cells and astrocytes[42].

The AUC of the protection model with 5 proteins (ISLR2, MYOC, WFIKKN1, SCARA5, and ADAM15) that distinguished nAMS1k and nAMS4k was 0.95 ($p$ value < 0.001, Fig. 3d), which indicated that these proteins may be involved in attenuating the symptoms of AMS in individuals without AMS at high altitudes. The protein-level responses of individuals with and without AMS to high altitude were notably different. Notably, ADAM15 was identified as an essential candidate protective protein in the differential abundance analysis (Fig. 3a) and could therefore be a target for preventing and treating AMS.

The AUC of the prediction model distinguishing AMS1k and nAMS1k with 8 proteins (PHGDH, ubiquitin like modifier activating enzyme 1 (UBA1), RBKS, G protein subunit alpha 13 (GNA13), insulin like growth factor binding protein 7 (IGFBP7), ficolin 2 (FCN2), CA2, and V-set and immunoglobulin domain containing 4 (VSIG4)) was 0.91 ($p$ value = 0.001, Fig. 3e). PHGDH and RBKS were also identified as candidate predictive biomarkers via differential abundance analysis, which suggested that these proteins are likely predictive biomarkers to evaluate the occurrence of AMS in individuals before their ascension to high altitude. In particular, PHGDH, which is involved in glucose/energy metabolism, had a fourfold higher weight than other proteins in the model.

The AUC of the diagnosis model distinguishing AMS4k and nAMS4k with 13 proteins (TRAF2, angiotensinogen (AGT), IL18R1, Thy-1 cell surface antigen (THY1), ISLR2, transcobalamin 2 (TCN2), CD38, G protein subunit alpha 14 (GNA14), INHBC, GC, RBKS, WFIKKN1, and ARF6) was 0.89 ($p$ value < 0.01, Fig. 3f), which indicated the potential translational value of these proteins as diagnostic biomarkers of AMS. TRAF2, AGT, and IL18R1 were the top 3 weighted proteins in this model. TRAF2, GC, and WFIKKN1 were also identified as candidate diagnostic biomarkers via differential analysis. TRAF2, a TNF receptor-associated factor, is associated with signal transduction

from members of the TNF receptor superfamily and apoptosis[43]. AGT, which is expressed in the liver, is involved in the maintenance of blood pressure, body fluid, and electrolyte homeostasis. IL18R1 is the receptor of the proinflammatory cytokine IL18. GC, a transporter of plasma metabolites binding to vitamin D, is reduced under hypoxia[44], which is consistent with our result that GC was downregulated in both individuals with AMS and individuals without AMS after ascending to high altitude.

Accordingly, we built robust models that distinguished these groups well and provided a panel of candidate biomarkers with promising translational value for each scenario.

**Key clinical indexes of AMS**. The identification of key clinical indexes affected by AMS could shed light on our understanding and redefinition of AMS. Overall, 18 clinical indexes, such as antibodies to thyroid peroxidase (A-TPO), albumin, complement C3c (C3C), free triiodothyronine (FT3), and lipase (Supplementary Fig. 7a, b), showed significant differences between the AMS4k and AMS1k groups and between the nAMS4k and nAMS1k groups (Fig. 3b, Supplementary Data 5). Among these indexes, lipase, a-Amylase EPS pancreatic (AMY-P), and carbohydrate antigen 19-9 (CA199) were downregulated (Supplementary Fig. 8a). In addition, C-peptide, creatine kinase-MB (CKMB), estradiol-E2 (E2), phosphate, progastrin-releasing peptide (proGRP), procalcitonin, thyroglobulin, and thyroxine showed significant differences between the AMS4k and AMS1k groups but not between the nAMS4k and nAMS1k groups, which reveals that these clinical indexes are affected by AMS (Fig. 3b). Bilirubin total DPD (BILT), cancer antigen 125 (CA125), ferritin, follicle-stimulating hormone (FSH), low-density lipoprotein (LDL), and prolactin showed significant differences between the nAMS4k and nAMS1k groups but not between the AMS4k and AMS1k groups (Fig. 3b, Supplementary Fig. 7b, Supplementary Data 5). Notably, BILT was upregulated in individuals who stayed at a high altitude for 1 month[45]. C3C, FT3, and total procollagen type 1 amino-terminal propeptide (TP1NP) show significant differences between the Han and Tibetan populations[46].

BILT, glucose, N-terminal pro B-type natriuretic peptide (proBNP), and thyrotropin were found to show significant differences between the AMS1k and nAMS1k groups (Supplementary Figs. 7c, 8c). TP1NP was significantly downregulated in the AMS4k group compared with the nAMS4k group, whereas C-peptide, carcinoembryonic antigen (CEA), γ-glutamyltransferase (GGT), insulin, and procalcitonin were significantly upregulated (Supplementary Figs. 7d, 8d). In addition, C-peptide, CEA, GGT, and procalcitonin were also significantly upregulated in the AMS4k group compared with the AMS1k group (Fig. 3b). The same trends between the two comparisons indicated the potentially remarkable roles of C-peptide, CEA, GGT, and procalcitonin in the pathogenesis of AMS. Procalcitonin is an inflammatory biomarker that is found at extremely low levels in the periphery of healthy individuals but is increased by inflammatory mediators[47], which suggests the probable role of inflammation in the occurrence of AMS. Collectively, the association between AMS and multiple clinical indexes indicates that AMS is a complex and systemic disease.

We built robust models to distinguish AMS4k from AMS1k, nAMS4k from nAMS1k, AMS4k from nAMS4k, and AMS1k from nAMS1k using the clinical indexes identified from each differential analysis by 10-fold cross-validation. We found very high classification accuracy (AUC = 0.96, $p$ value < 0.0001) in the pathogenesis model distinguishing AMS1k and AMS4k with 19 clinical indexes (such as A-TPO, C-peptide, phosphate, E2, CKMB, and thyroxine; Fig. 3c, Supplementary Fig. 9a). The AUC

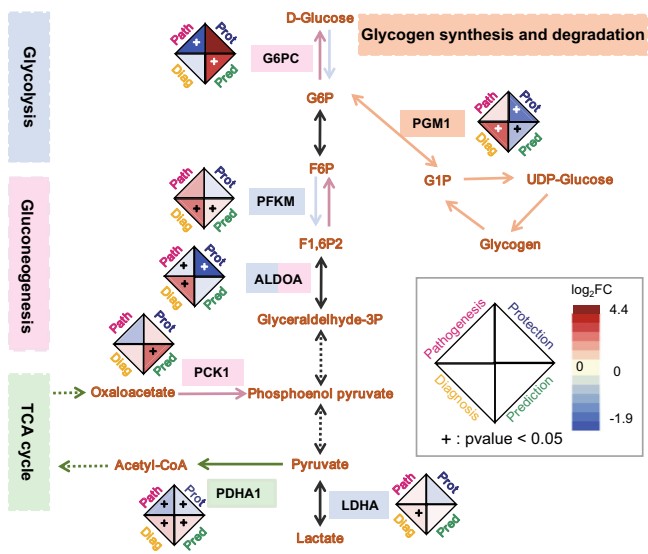

**Fig. 4 Changes in proteins involved in carbohydrate metabolism in AMS.** The log$_2$FC of proteins in glycolysis (light purple background and arrows), gluconeogenesis (light pink background and arrows), the TCA cycle (light green background and arrows), and glycogen synthesis and degradation (light orange background and arrows) in the following four comparison groups are shown in the diamond shape: pathogenesis (path, pink background, AMS4k: $n = 30$, AMS1k: $n = 30$), protection (prot, dark blue background, nAMS4k: $n = 23$, nAMS1k: $n = 23$), prediction (pred, dark green background, AMS1k: $n = 30$, nAMS1k: $n = 23$), and diagnosis (diag, yellow background, AMS4k: $n = 30$, nAMS4k: $n = 23$). Carbohydrate metabolism was dysregulated in individuals after ascension to high altitude. The statistical methods were used paired/unpaired $t$ test or Welch's $t$ test. TCA tricarboxylic acid cycle.

of the protection model distinguishing nAMS1k and nAMS4k with a panel of 16 clinical indexes (such as FT3, ferritin, free Prostate-specific antigen (FPSA), creatinine, C3C, and FSH) was 0.94 ($p$ value < 0.001, Fig. 3d, Supplementary Fig. 9a), and the AUCs of the prediction and diagnosis models distinguishing AMS1k and nAMS1k with 3 clinical indexes (proBNP, BILT, and thyrotropin) and ASM4k and nAMS4k with 4 clinical indexes (procalcitonin, TP1NP, C-peptide, and insulin) were 0.67 and 0.7, respectively ($p$ values > 0.05, Fig. 3e, f). It appears that clinical indexes showed weaker performance than proteins for the prediction and diagnosis of AMS. A-TPO, which exhibited the highest weight in the pathogenesis comparison but not in the protection comparison (Supplementary Fig. 9a, b), is reportedly higher in patients with hyperemesis gravidarum than in nonpregnant controls[48], which indicates its association with vomiting.

Models were also established using a combination of clinical indexes and protein data to explore candidate biomarkers for AMS. The AUCs of the pathogenesis and protection models obtained using proteins and clinical indexes were 0.98 ($p$ value < 0.00001, Fig. 3c) and 0.9 ($p$ value < 0.01, Fig. 3d) between AMS4k and AMS1k and nAMS4k and nAMS1k, respectively. A-TPO continued to exhibit the highest weight between the AMS4k and AMS1k groups, whereas RET, MYOC, and MATN3 were also selected from the combined data (Supplementary Fig. 9b). The classification performances of the pathogenesis (Fig. 3c) and protection (Fig. 3d) models obtained with proteins alone were comparable to those of the models obtained with the combined data.

In the prediction and diagnosis models, the classification performance obtained with protein alone and with the combined

data was similar. Notably, fewer features remained in the prediction model obtained with proteins alone than in those obtained with the combined data (Fig. 3e, f, Supplementary Fig. 9b). In addition, all important features remaining in the diagnosis model obtained with the combined data were proteins. Accordingly, we hypothesized that proteins are better choices than clinical indexes for the prediction and diagnosis of AMS.

**Changes in carbohydrate metabolism between individuals with and without AMS.** Based on the dysregulation of PFKM and RBKS identified by PEA (Fig. 3a), we extended the protein panel used in the MRM assay to comprehensively profile energy metabolism, particularly gluconeogenesis, glycolysis, and the tricarboxylic acid cycle (TCA cycle) (Fig. 4, Supplementary Fig. 10a).

Individuals with AMS showed higher utilization of gluconeogenesis than individuals without AMS at low altitude. The two key enzymes of gluconeogenesis, glucose-6-phosphatase catalytic subunit (G6PC) and PCK1, presented higher abundance in the AMS1k group than in the nAMS1k group (Fig. 4 and Supplementary Fig. 10a). Notably, G6PC is a subset of glucose-6-phosphatase catalyzing the hydrolysis of D-glucose 6-phosphate (G6P) to D-glucose, and PCK1 is a rate-limiting enzyme of gluconeogenesis.

Individuals with AMS may exhibit higher utilization of glycolysis than individuals without AMS at high altitude, and this hypothesis is supported by the following two points. First, PFKM, a key rate-limiting enzyme in glycolysis, was found at a higher level in AMS4k than in nAMS4k (Fig. 4 and Supplementary Fig. 10a), which would lead to the production of more fructose 1,6-bisphosphate and thereby the stimulation of glycolysis. Second, individuals with AMS showed higher levels of lactate dehydrogenase A (LDHA) and ALDOA than individuals without AMS at high altitude. LDHA and ALDOA are involved in both glycolysis and gluconeogenesis. However, the gluconeogenesis-related enzymes PCK1 and G6PC showed similar expression levels between AMS4k and nAMS4k (Supplementary Fig. 10a). Therefore, we hypothesized that glycolysis, rather than gluconeogenesis, was more active in the AMS4k group than in the nAMS4k group.

Individuals with AMS may have a more active TCA cycle than individuals without AMS at low altitude and at high altitude, although inhibition of the TCA cycle was found in both groups of individuals after ascension to high altitude (Fig. 4). PDHA1, a subset of pyruvate dehydrogenase, was downregulated after ascension to high altitude in both individuals with AMS and those without AMS. The downregulation of PDHA1 inhibited the conversion between pyruvate and acetyl-CoA and thereby downregulated the TCA cycle (Supplementary Fig. 10a), but the level in individuals with AMS remained higher than that in individuals without AMS.

Individuals without AMS had a lower utilization of glycogen after ascension to high altitude to aid in acclimation, whereas individuals with AMS may consume more glycogen for glycolysis than individuals without AMS at high altitude. Both UDP-glucose and food-derived glucose can participate in glycogen synthesis[49]. Glycogen can degrade into glucose 1-phosphate (G1P). The reversible isomerization between G1P and glucose 6-phosphate (G6P) is catalyzed by phosphoglucomutase 1 (PGM1). PGM1 deficiency leads to a failure to utilize glycogen as an energy source in both the liver and skeletal muscle[50]. In our study, PGM1 expression was lower in nAMS4k than in nAMS1k, lower in AMS1k than in nAMS1k, and higher in AMS4k than in nAMS4k (Supplementary Fig. 10a).

In summary, individuals with AMS may exhibit stronger gluconeogenesis ability at low altitude and higher utilization of glycogen and glycolysis at high altitude than individuals without

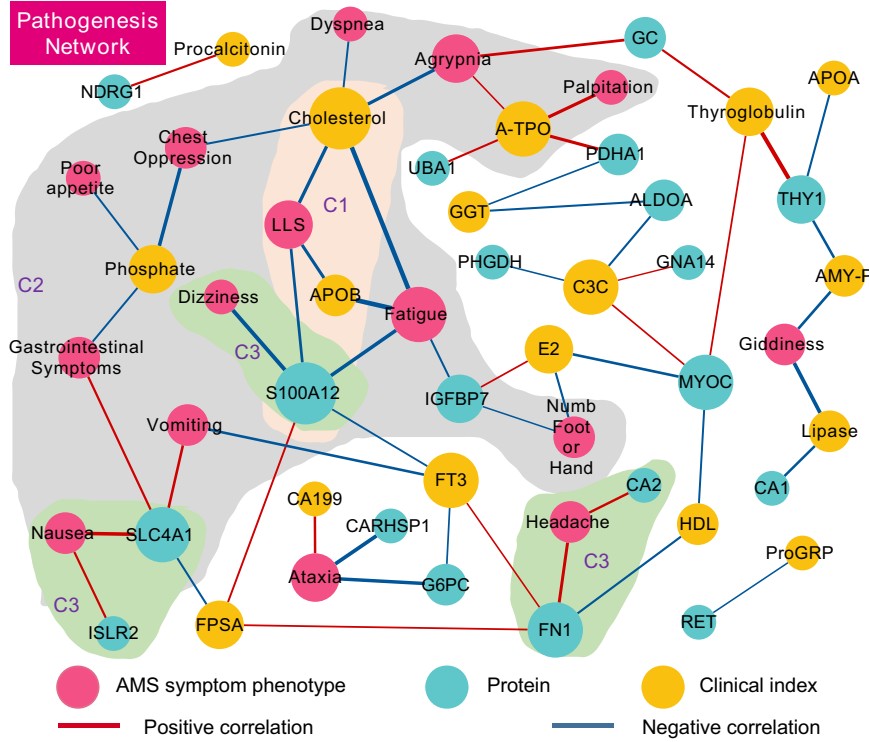

**Fig. 5 Network visualization of MRM-identified DEPs, clinical indexes and AMS symptom phenotypes in the pathogenesis comparison.** The network of AMS symptom phenotypes (pink dots), DEPs (blue dots), and clinical indexes (yellow dots) was connected based on Spearman correlation coefficients and statistical significance in 30 individuals with AMS. The edges showing a positive correlation (red line) and a negative correlation (blue line) with $p$ values < 0.05 are shown. The thickness of the edges corresponds to the absolute value of the correlation coefficients. The size of the node is proportional to the number of edges from the node. Cluster C1 (light orange background area), cluster C2 (light green), and cluster C3 (light gray) are labeled C1, C2, and C3 (purple fonts), respectively. S100A12 was found to be associated with the LLS of AMS and with several proteins, such as CA2 and SLC4A1, which suggests their potential for development as novel clinical indexes.

AMS. Individuals without AMS may inhibit the utilization of glycogen for storage after ascension to high altitude. After ascension to high altitude, all individuals showed a decreased TCA cycle, whereas individuals with AMS showed a more active TCA cycle than individuals without AMS. We argue that different carbohydrate metabolism pathways play a key role in acclimation to high altitude.

**Association among proteins, AMS symptom phenotypes, and clinical indexes.** Associating the 22 symptom phenotypes of AMS with the changes in key proteins and clinical indexes between the AMS4k and AMS1k groups (Fig. 5, Supplementary Data 6) could clarify the AMS symptom phenotypes at a molecular level and better redefine AMS using proteins and/or clinical indexes. A correlation analysis showed that LLS was negatively correlated with S100A12, apolipoprotein B (APOB), and cholesterol ($\rho s < -0.44$, $p$ value < 0.05) in the C1 cluster (Fig. 5). Furthermore, S100A12, SLC4A1, IGFBP7, APOB, A-TPO, phosphate, and cholesterol were correlated with multiple symptom phenotypes, such as fatigue, nausea, vomiting, and poor appetite (C2 in Fig. 5), which indicates their potential important roles in the development of AMS. Hypoxia together with elevated inorganic phosphate could enhance vascular smooth muscle cell osteogenic transdifferentiation[51]. SLC4A1 is associated with $CO_2$ gas transport in erythrocytes[52].

Several AMS symptom phenotypes were correlated with proteins but not with any clinical index (C3 in Fig. 5). For example, headache was positively correlated with CA2 and fibronectin 1 (FN1) ($\rho s > 0.42$, $p$ value < 0.05). Dizziness was negatively correlated with S100A12 ($\rho = -0.51$, $p$ value < 0.05), and nausea was positively correlated with ISLR2 and SLC4A1

($\rho s > 0.41$, $p$ value < 0.05). These results indicate opportunities for the development of novel clinical indexes based on these proteins to better redefine AMS.

The correlation analysis among clinical indexes, proteins, and AMS symptom phenotypes in 53 individuals at low altitude or at high altitude revealed that several features could contribute to the prediction, pathogenesis, or diagnosis of AMS. No significantly different features between AMS1k and nAMS1k were found to be correlated with the AMS symptom phenotypes (Supplementary Fig. 11a). However, a connection between thyrotropin and the glucose/energy metabolism-related proteins PCK1, PHGDH, and IGFBP7 and the phospholipid catabolism-related protein phospholipase A2 group VII (PLA2G7) was found between AMS1k and nAMS1k. The SNPs of thyrotropin and PLA2G7 are associated with blood pressure variations and hypertension, respectively[53]. Moreover, rather than clinical indexes, proteins, including FGF23, RET, IL18R1, and GNA14, were positively correlated with AMS symptom phenotypes, such as poor appetite, dyspnea, and lip cyanosis, between the AMS4k and nAMS4k groups (Supplementary Fig. 11b). In addition, C-peptide was positively correlated with 13 proteins, such as the candidate pathogenesis-related protein MYOC ($\rho = 0.42$, $q$ value < 0.05), the protective protein KITLG ($\rho = 0.27$, $q$ value < 0.05), and the diagnostic biomarker TRAF2 ($\rho = 0.36$, $q$ value < 0.05). Accordingly, C-peptide may be involved in the pathogenesis of AMS, whereas the clinical indexes investigated in our study did not perform better than proteins in the prediction and diagnosis of AMS.

**Discussion**
We performed a systematic study of AMS using plasma proteome and clinical indexes. We measured a total of 40,248

measurements including 21,380 measurements by PEA, 10,812 measurements by MRM, 6890 measurements of clinical indexes, and 1166 measurements of AMS symptom phenotypes. Specifically, we profiled 1069 proteins from 10 individuals with AMS at low altitude and at high altitude using Olink's PEA technology and identified 47 AMS-relevant proteins. PEA technology exhibits high sensitivity down to concentrations in the pg/mL range, which allows the detection of low-abundance proteins[54]. We validated 102 proteins using MS-based MRM technology in 53 individuals with or without AMS at low altitude and at high altitude. The combination of PEA and MRM could largely avoid false positives, increasing the quality of the candidate biomarkers[54]. We then systematically analyzed the proteins and clinical indexes that may be involved in the pathogenesis, protection, prediction, and diagnosis of AMS to identify candidate therapeutic targets and biomarkers. We also discovered carbohydrate metabolism dysregulation in individuals with AMS and without AMS after ascension to high altitude. Moreover, we profiled 22 AMS symptom phenotypes and 65 clinical indexes in the same cohort and identified strong correlations among AMS-related proteins, phenotypes, and clinical indexes, which provides a basis for the precise redefinition of AMS using proteins and clinical indexes.

The PEA and MRM assays confirmed that plasma RET was significantly increased in individuals with AMS at high altitude compared with that found in these individuals at low altitude (Supplementary Fig. 10b). However, RET was decreased in individuals without AMS at high altitude compared with the levels found in these individuals at low altitude (Supplementary Fig. 10b). RET was identified as the most important variable in the machine learning-based model, with high classifier accuracy (panel, AUC = 0.9; single, AUC = 0.74) for the AMS4k and AMS1k groups (Fig. 3c, Supplementary Fig. 9c).

RET is correlated with a basic symptom of AMS. RET was previously shown to regulate the survival and size of nociceptors that transmit information to the brain, leading to the sensation of pain[55,56]. In our study, RET was positively correlated with headache ($\rho = 0.35$, $p$ value = 0.066, Supplementary Data 6). This result suggests that RET is likely involved in the pathophysiological mechanism of headache, a key phenotype of AMS.

RET could increase the promoter activity of CA9 induced by HIF-1α under hypoxia[57,58]. CA9, which is homologous with CA1, CA2, and CA6, could maintain the homeostasis of the blood oxygen partial pressure and acid-base balance by regulating ventilation under hypoxia[59,60]. CA2, which was also selected by XGBoost, was identified as a candidate predictive biomarker for AMS in our study. Acetazolamide, an inhibitor of CA2 and CA9, is a class of drugs to prevent and treat AMS. These results suggest that RET likely also interacts with CA2. In addition, ISLR2, which exhibited the top weight in the machine-learning model to distinguish nAMS4k and nAMS1k, is related to RET[41]. Remarkably, selective inhibitors of RET, selpercatinib and pralsetinib, were recently approved for the treatment of RET-associated non-small-cell lung cancer[61,62]. In summary, RET could be a therapeutic target in the treatment of AMS.

In addition, proinflammatory programs are selectively activated by TRAF2 and TNF receptor-associated factor 6 (TRAF6), which are associated with RET/PTC oncoproteins in papillary thyroid carcinoma[63]. TRAF2 was also selected by both differential analysis and XGBoost between AMS4k and nAMS4k as a candidate diagnostic biomarker. In addition, RET was positively correlated with lip cyanosis, an AMS symptom phenotype. Therefore, RET may be an essential diagnostic biomarker for AMS.

TRAF2 was selected via differential analysis and the XGBoost classifier, with an AUC of 0.89 between AMS4k and nAMS4k, and 0.84 from single protein (Fig. 3f, Supplementary Fig. 9c). TRAF2

interacts with procaspase-12 and promotes the activation of caspase-12[64], which can transduce signals from inositol-requiring enzyme 1 (IRE1s) under endoplasmic reticulum (ER) stress conditions and lead to apoptosis[65]. Moreover, salidroside reportedly reduces TRAF2 to protect against hypoxia-induced liver injury by inhibiting ER stress-mediated apoptosis[66]. Accordingly, TRAF2 was identified as a candidate diagnostic biomarker for AMS, and salidroside was found to be a candidate drug for the treatment of AMS.

The PEA and MRM assays confirmed that S100A12 was significantly decreased in individuals with AMS at high altitude compared with the levels found in these individuals at low altitude (Fig. 3a). Among all the groups, the highest expression of S100A12 was found in the AMS1k group (Supplementary Fig. 10b). In addition, S100A12 was one of the key features in the machine learning-based pathogenesis model, with high accuracy, and was identified as a candidate predictive biomarker via differential analysis between the AMS1k and nASM1k groups. Moreover, S100A12 was negatively correlated with LLS, dizziness and fatigue. These findings indicate that S100A12 is highly involved in the pathogenesis of AMS.

The downregulation of S100A12 did not inhibit the inflammatory response induced by hypoxia. The inflammatory response was actively involved in AMS, as shown previously. The proinflammatory factor S100A12, a type of endogenous innate danger molecule, could provoke proinflammatory responses in endothelial cells[67]. S100A12 levels are associated with increased levels of markers of pulmonary inflammation and hypoxia in patients undergoing cardiac surgery[68]. In addition, the downregulation of S100A12 in aortic smooth muscle could reduce apoptosis[69] but had no significant effect on inflammatory signaling in monocytes[70]. Therefore, it appears that the downregulation of S100A12 likely reduces apoptosis but does not inhibit the inflammation induced by hypoxia in individuals with AMS after ascension to high altitude.

ADAM15 was identified as a candidate protective biomarker of individuals without AMS and was selected by the machine learning-based model, with high accuracy (panel, AUC = 0.95; single, AUC = 0.80) in distinguishing nAMS1k and nAMS4k (Fig. 3d, Supplementary Fig. 9c). ADAM15 is involved in the response to hypoxia, proteolytic ectodomain processing of cytokines, cell adhesion signaling, and angiogenesis in endothelial cells[71,72], and it is associated with atherosclerosis, rheumatoid arthritis, intestinal inflammation, and inherent angiogenesis[73]. In addition, the silencing of ADAM15 can inhibit the expression of proinflammatory cytokines in rheumatoid angiogenesis[72]. Moreover, systemic proinflammatory cytokines are associated with the development of AMS and HAPE[74,75]. Proteins involved in the inflammatory response, such as FGF23, KITLG, and plasminogen activator, urokinase (PLAU), were found at lower levels in individuals without AMS than in those with AMS, but CCL2 was not significant in any of the compared groups (Supplementary Fig. 10b). This finding is consistent with the fact that CCL2 is independent of AMS status[11]. Importantly, adequate anti-inflammatory properties favor resistance to AMS[11]. The candidate pathogenesis biomarker RET, which was higher in individuals with AMS than in those without AMS, was associated with proinflammation. In conclusion, subjects without AMS may rapidly acclimate to a high-altitude environment by downregulating ADAM15 and inflammation. Thus, ADAM15 could be a target in the treatment of AMS.

PHGDH, which was the top-weighted protein identified by the XGBoost classifier, with an AUC of 0.91 between AMS1k and nAMS1k from the panel and 0.84 from single protein, was identified as a candidate predictive biomarker of AMS (Fig. 3d, Supplementary Fig. 9c). PHGDH is involved in glucose and energy metabolism. The inhibition of PHGDH could downregulate NADPH levels, disorder mitochondrial redox

homeostasis, and increase apoptosis under hypoxia[76,77]. The overexpression of PHGDH reduces hypoxia-induced cell death[77]. Moreover, in our study, PHGDH was upregulated in individuals without AMS and downregulated in individuals with AMS to a similar level after ascension to high altitude (Supplementary Fig. 10b). Collectively, the results indicate that PHGDH may be a promising predictive biomarker for AMS.

C-peptide was selected via both differential analysis and the classifier between the AMS4k and AMS1k and the AMS4k and nAMS4k groups, with AUCs greater than 0.7, but not between the nAMS4k and nAMS1k groups. C-peptide, which is a polypeptide that connects two chains of proinsulin, was upregulated in AMS4k compared with AMS1k and was higher in individuals with AMS than in those without AMS at high altitude. In addition, our study showed that C-peptide was positively correlated with proteins related to the pathogenesis, protection and diagnosis of AMS. Moreover, C-peptide could increase proliferation[78] and activate anti-inflammation in endothelial cells[79]. C-peptide was reportedly elevated in 7 subjects with AMS after ascension to high altitude[80], which is consistent with our result (Fig. 3b). However, an acclimatization study revealed that C-peptide was significantly decreased at 3600 m compared with sea level, but no significant difference was found between sea level and an altitude of 4650 m and above, regardless of the AMS status[81]. In our study, C-peptide showed the upregulation trend in both individuals with and without AMS at high altitude compared with the levels found in these individuals at low altitude. Taken together, the results show that C-peptide may be associated with the pathogenesis and diagnosis of AMS.

Individuals with AMS may have a more active TCA cycle than individuals without AMS in response to hypoxia and show enhanced glycolysis and increased utilization of glycogen compared with individuals without AMS at high altitude. In addition, individuals with AMS exhibit active gluconeogenesis at low altitude. Regardless of whether the individuals suffered from AMS at high altitudes, individuals at high altitudes showed a reduced TCA cycle due to the hypoxic environment. Based on the downregulation of TCA-related enzymes (such as PDHA1) and glycolysis-related enzymes (such as ALDOA) in the AMS-resistant group, Lu et al.[15] reported that the TCA cycle and glycolysis are reduced in individuals without AMS but not in individuals with AMS after exposure to high altitude. It appears that the balance between glycolysis and gluconeogenesis is relevant to AMS. These differences could aggravate the consumption of oxygen, leading to discomfort in individuals with AMS and comfort in individuals without AMS at high altitude. Additionally, an enzyme related to glycogenesis, PCK1, was identified as a candidate predictive biomarker for AMS in our study, and this finding highlights the potentially important role of gluconeogenesis in the prediction of AMS.

We built four robust machine-learning models to dissect the pathogenesis of AMS, screen therapeutic targets and identify protective, predictive and diagnostic biomarkers. Using only several proteins, these models maintained high accuracy (AUCs ≥ 0.9), which indicates the practical value of these models. In particular, we could screen individuals susceptible to AMS using predictive biomarkers to largely prevent the occurrence of AMS, which is undesirable to individuals who would like to ascend to high altitude. The pathogenesis and protection models obtained using clinical indexes exhibited high accuracy, whereas the prediction and diagnosis models established using clinical indexes did not perform as well as those obtained using proteins. Moreover, the prediction and diagnosis models obtained using both clinical indexes and proteins did not perform as well as those established using only proteins, and this finding indicates that these proteins are more suitable for the prediction and

diagnosis of AMS than these clinical indexes, which shows the potency of the identified proteins as novel clinical indexes.

The redefinition of AMS based on proteins and clinical indexes will promote an improved understanding and precise treatment of AMS. Currently, the diagnosis of AMS mainly depends on the self-questionnaire LLS. Here, we propose a panel of candidate predictive biomarkers (such as PHGDH, UBA1, RBKS, GNA13, IGFBP7, CA2, and VSIG4) and a panel of candidate diagnostic biomarkers (such as TRAF2, AGT, IL18R1, ISLR2, GC, RBKS, and WFIKKN1) for AMS. In addition, C-peptide could be an assistant diagnostic biomarker for AMS identified via differential analysis, the machine-learning model, and correlation analysis with multiple AMS-relevant proteins.

In this study, we used 106 plasma samples paired from 53 individuals with robust statistical and machine learning-based models to comprehensively profile AMS utilizing PEA and MRM-based proteomic technology. To the best of our knowledge, this is the largest cohort used in a study of AMS Notably, the evidence we provide represents a combination of molecular evidence on the mechanistic roles of these proteins, and the direct observation of biological changes upon intervention in our longitudinal cohort. We plan to further validate these biomarkers using another independent cohort, covering a larger number of individuals, including women, other races, other age ranges, and explore the association between AMS and other phenotypes, such as psychological factors, to contribute to translational medicine.

We systematically profiled the characteristics of AMS using two proteome technologies based on different principles, PEA and MRM, and 106 plasma samples. We validated that RET actively participates in the pathogenesis of AMS and is a candidate therapeutic target. The downregulation of ADAM15 may play a role in preventing individuals from developing AMS. PHGDH is a candidate predictive biomarker, and TRAF2 is a promising diagnostic biomarker. Furthermore, we built robust machine learning-based diagnosis, prognosis, protection, and pathogenesis models with high classification accuracy and thereby validated the roles of these proteins in AMS. Individuals with AMS may exhibit more active gluconeogenesis at low altitude than individuals without AMS and enhanced utilization of glycogen compared with individuals without AMS at high altitude. Additionally, we profiled the associations among 22 symptom phenotypes of AMS, 65 clinical indexes, and these proteins. Our findings shed light on redefining AMS based on proteomic and clinical biomarkers, instead of a self-questionnaire, contributing to precision medicine of AMS and improving our understanding of acclimatization to extreme environments.

## Methods

**Subjects.** A total of 53 Han Chinese male subjects (aged 18–20 years) were recruited in this study. The exclusion criteria included having any health problems; having any known liver, lung, or cardiovascular disease; a history of migraine or head injury; smoking; and having been to altitudes >2500 m or exposed to a hypobaric hypoxic environment within the last 3 months. Ethical approval was obtained from the Chinese PLA General Hospital ethical committee with the approval identifier S2019-035-01, and all protocols followed the established national and institutional ethical guidelines. All participants provided signed written informed consent.

**Evaluation of the AMS status at high altitude.** The AMS status of 53 subjects was evaluated according to the LLS[4], a self-reported scoring standard, after ascension to high altitude (4300 m). Briefly, a four-point scale (asymptomatic = 0, mild = 1, moderate = 2, severe = 3) was used to quantify the degree of headache, gastrointestinal symptoms (poor appetite, nausea/vomiting), fatigue and dizziness (Supplementary Data 1). Subjects with severe headache but no other symptoms of AMS or an LLS greater than 2 were defined as individuals with AMS (AMS, $n = 30$). Subjects with an LLS <3 and subjects without headaches were defined as individuals without AMS (nAMS, $n = 23$). With the exception of the 7 symptom phenotypes used for calculating the LLS, 14 other AMS symptom phenotypes were also evaluated for further analysis (Supplementary Data 1).

**Experimental setup, sample collection, and biochemical detection.** All subjects were transported to an altitude of 4300 m (4 km) from 1200 m (1 km) within 4 h by plane. The AMS symptoms were evaluated after ascension to 4 km. Peripheral venous whole-blood samples of 53 subjects (30 AMS and 23 nAMS) were collected at an altitude of 1 km (AMS1k and nAMS1k) and after arrival at an altitude of 4 km (AMS4k and nAMS4k) for 1–4 days. Blood samples were collected in a semi-recumbent position from an anterior elbow vein by conventional venipuncture and placed in an EDTA-coated blood collection tube. Plasma was separated by centrifugation and stored in a 0.5 mL aliquot at −80 °C until analysis. All samples from these subjects at both time points were also prepared for biochemical detection. Two milliliters of each plasma sample was used to assay 65 clinical indexes (Supplementary Data 1) using a hematology analyzer (cobas 6000; Roche, USA).

Twenty paired plasma samples from 10 individuals with AMS at 1 km (AMS1k) and 4 km (AMS4k) were selected to identify the protein expression profile using Olink's PEA technology. Subsequently, 60 paired plasma samples from 30 individuals with AMS (including the subjects used for PEA) and 46 paired plasma samples from 23 individuals without AMS were used for validation.

**Plasma proteome profiling and analysis.** We analyzed 20 plasma samples from individuals with AMS using Olink's PEA technology with the remaining 12 commercially available panels at that time, excluding 1 panel with overlap in most proteins, by iCarbonX (Shenzhen) Company Limited. These panels consisted of 5 disease panels (Cardiovascular II panel, Cardiovascular III panel, Inflammation panel, Neurology panel, and Oncology II panel) and 7 important biological progress panels (Neuro Exploratory panel, Development panel, Cardiometabolic panel, Immune Response panel, Cell Regulation panel, Metabolism panel, and Organ Damage panel), as indicated on the manufacturer's website (Olink Proteomics, Uppsala, Sweden). For each panel, each serum sample (1 μL) was added to 3 μL of the incubation mix and incubated at 4 °C overnight (16–22 h). The extension mix was prepared by mixing PEA enzyme and PCR reagents in nuclease-free water. A total of 96 μL of extension mix was added to the samples and immediately transferred to the thermal cycler, allowing a 20-min DNA extension at 50 °C, followed by 17 cycles of DNA amplification. Further, 2.8 μL of post-PCR product was mixed with 7.2 μL of detection mix containing PCR polymerase and real-time PCR reagents in a new 96-well plate. The mixture and PCR primers were loaded onto the primed microfluidic chip (96.96 Dynamic Array IFC, Fluidigm, USA), followed by real-time PCR performed in the Biomark HD system (Fluidigm, USA) using the program provided by Olink Biosciences. The protein concentrations were finally normalized and transformed using internal and interplate controls to adjust for intra- and inter-run variation[82]. More detailed information can be found in the panel-specific validation data documents (www.olink.com/downloads). The expression levels of proteins were represented as linear Normalized Protein eXpression (NPX), a relative quantification scale in arbitrary units. A complete list of all 1,104 measured proteins (1069 unique proteins) can be found in the Supplementary Data (Supplementary Data 2).

Proteins with coefficients of variation <0.3 and a missing data frequency less than 0.25 were used for further analysis. The values of undetected features were replaced with 1/10 of the minimum nonzero value[83]. Paired two-tailed $t$ test or paired two-tailed Welch's $t$ test were performed for statistical analyses based on homoscedasticity, and the $p$ values were corrected using Bonferroni-Hochberg (BH) corrections ($q$ value) for multiple comparisons. Proteins with $q$ values <0.05 were considered DEPs. PCA was performed using the DEPs without missing values via the package ggplot2[84]. The heatmap of DEPs was generated using the R package heatmaply[85] with scaled raw data.

**DEP validation by multiple reaction monitoring.** MRM was applied to verify the 47 selected DEPs measured by PEA and 55 other pathway-related or interest proteins at the validation stage (Supplementary Data 2), which consisted of 106 samples (30 individuals with AMS and 23 nAMS at 1 km and 4 km, respectively, involving the 10 paired samples measured by PEA) by Beijing Qinglian Biotech Company Limited. A total of 538 transitions were selected to represent the 102 proteins. The unique peptide and transition for MRM from the peptides identified by TripleTOF 5600+ mass spectrometry (SCIEX) in the mixed plasma samples were selected using the Skyline software from a background database of human species, and further screened with the transition library in SRMatlas (https://db.systemsbiology.net/sbeams/cgi/PeptideAtlas/GetTransitions). Three transitions were selected for each peptide, and two peptides were kept for each protein. The MRM assay was performed using a QTRAP 6500+ mass spectrometer (SCIEX), and the quantitative method was constructed by the Analytics module in SCIEX OS software (version 2.0).

A total of 10 μL of plasma samples were subjected to reduction. We added 20 mM dithiothreitol (DTT) solution and incubated at 37 °C for 1 h. Subsequently alkylation was performed with sufficient iodoacetamide (IAM) for 1 h at room temperature in the dark. The sample was diluted 4 times by adding a 25 mM ammonium bicarbonate (ABC) buffer. Then, trypsin (trypsin:protein = 1:50) was added and incubated at 37 °C overnight. The next day, 50 μL of 0.1% FA was added to terminate the digestion. Finally, the samples were all desalted with a C18 cartridge to remove the high urea, and desalted samples were dried by vacuum centrifugation for MRM analyses.

By using BSA peptides for normalization, we obtained the standardized abundance value (intensity) and performed relative quantitation according to grouping. Transitions with missing value frequencies greater than 25% were removed. Further calculations and two-tailed statistical tests were conducted using paired $t$ test or paired Welch's $t$ test for the self-control compared groups and $t$ test or Welch's $t$ test for the other compared groups in R based on homoscedasticity. The $p$ values were corrected using BH ($q$ value) for multiple comparisons. Proteins with $q$ values < 0.05 were considered DEPs. The heatmap of the intersection of MRM-validated DEPs selected via differential analysis in four compared groups was generated using the pheatmap package, and the violin plot was generated using ggplot2.

**Functional enrichment analysis.** Gene Ontology biological process (GOBP) and Kyoto Encyclopedia of Genes and Genomes (KEGG) enrichment analyses of the DPEs measured by PEA and MRM were performed using the Bioconductor R package clusterProfiler[86]. The redundant GO terms were removed using the simplifying function (by = "p.adjust", cutoff = 0.3). The statistical significance of the GO enrichment was tested using Benjamini and Hochberg with a cutoff $q$ value < 0.05. All the KEGG analyses with $p$ values < 0.05 were enriched and shown.

**Machine learning-based models.** XGBoost, a boosted ensemble algorithm, was implemented, and tenfold cross-validation was performed using MRM-based proteomic data and/or clinical indexes for the identification of biomarkers. The proteins used for prediction were selected at the validation stage based on a $q$ value < 0.05. Clinical indexes were selected based on $q$ values < 0.05 between the AMS4k and AMS1k groups and the nAMS4k and nAMS1k groups and by $p$ values < 0.05 between the AMS1k and nAMS1k groups and the AMS4k and nAMS4k groups. The datasets between each comparison were divided into a training set (60%) and a test set (40%). XGBoost was performed using the caret package with the xgbTree method. Receiver operating characteristic (ROC) curves were generated to assess the AUC with the pROC package[87].

**Correlations among proteins, clinical indexes, and symptom phenotypes.** Spearman correlation analyses among proteins, clinical indexes, and AMS symptom phenotypes were performed using the psych package, and the results were visualized using Cytoscape software[88]. Samples with <75% of observations were eliminated during the correlation analysis. In addition, proteins and clinical indexes observed in less than 75% of the samples were deleted, while symptom phenotypes observed in less than 25% of the samples were deleted with the exception of the symptom phenotypes involved in LLS between the AMS4k and AMS1k groups as well as the AMS4k and nAMS4k groups. Features with a difference in detection rate >50% between the two groups were retained. Additionally, features with the same value in each sample were deleted. The $p$ values were corrected using BH ($q$ value) for multiple comparisons. Connections with $p$ values < 0.05 were selected for network visualization between the AMS4k and AMS1k groups and the nAMS4k and nAMS1k groups because no connections showed $q$ values < 0.05, whereas connections with $q$ values < 0.05 were selected from the other two compared groups.

**Statistics and reproducibility.** For the baseline data, continuous variables are presented as the means ± standard deviations (SDs) and medians with interquartile ranges (IQRs), and the ordinal and nominal variables are presented as percentages. The NPX data generated from PEA were compared by paired $t$ test. The data generated by MRM were compared by paired and unpaired two-tailed $t$ test or Welch's $t$ test based on homoscedasticity. The clinical indexes were compared by two-tailed $t$ test, Welch's $t$ test, or Wilcoxon test (paired or unpaired) considering a normal distribution and homoscedasticity. The $p$ values were corrected using BH ($q$ value) for multiple comparisons. Proteins with $q$ values < 0.05 were defined as DEPs. All statistical analyses were performed using R (version 4.0.2).

**Ethics approval.** Ethical approval was achieved from the Chinese PLA General Hospital ethical committee with the approval identifier S2019-035-01, and all protocols followed the established national and institutional ethical guidelines. All participants provided signed written informed consent.

**Reporting summary.** Further information on research design is available in the Nature Research Reporting Summary linked to this article.

## Data availability
The raw MRM proteomic data analyzed in this study are available at iProX[89] with the corresponding dataset identifier PXD029063.

## Code availability
The source code is freely available at Zenodo[90] and Github (https://github.com/Monica1227/AMS_biomarker).

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

## Acknowledgements

We thank the participants who donated samples and all colleagues who contributed to this study during the sample collection. This work was supported by the National Natural Science Foundation of China [31701155] and the Science and Technology Innovation Special Zone [19-163-12-ZD-037-003-02].

## Author contributions

Conceptualization and Supervision: K.H., Z.J.; Sample collection: J.S., X.Z. and Z.J.; Data analysis: J.Y., X.S.; Data interpretation: Z.J. and J.Y.; Writing: J.Y.; Editing: Z.J., J.Y. and K.H.; Revising: J.Y., Z.J., and X.W. All authors read and approved the final paper.

## Competing interests

The authors declare no competing interests.
