## [Peer Review File · Communications Biology]

Reviewer #2 (Remarks to the Author):

Dear Authors,

This is an interesting study in the direction of searching for the protective, predictive, and diagnostic biomarker as well as for understanding the pathogenesis. I would like to thank the authors for such an elaborative and holistic study that may be useful for understanding the disease better. However, there are small queries and suggestions that if included, might be helpful for readers.

Here are few comments and suggestions:

1. Give full-explanation of abbreviations at the first appearance such as RET, LLS and more.
2. In line 40: "providing an ideal model". Any word like "ideal", "first time" are big claims and should be avoided.
3. In lines where some facts are stated, an apt reference is required, such as –
 - a. Lines 43- 44 – Definition of AMS with an altitude of above 2500 m
 - b. Line 49 – "multi-factors, such as race, age..." reference for race and age can be mentioned if different from altitude attained reference.
 - c. Line 197 – "MYOC, is involved in cytoskeletal function"
 - d. Line 236 – "CD38 and KITLG are involved in the hematopoietic cell lineage"
 - e. Line 245 and 246 – "Among these DEPs, PCK1 and RBKS are involved in the metabolism of carbohydrates."
 - f. Line 291 – "TRAF2, a TNF receptor-associated factor, is associated with signal transduction from members of the TNF receptor superfamily and apoptosis"
 - g. In Line 435 – "such as poor appetite, dyspnea, and lip cyanosis"
4. In line 72: authors have mentioned that ANGPTL4 and resistin were associated with AMS, how are they associated? They promoting or helping in suppress the sickness?
5. The sample cohort is a good number however, it is a bit confusing the way it is mentioned. If authors can be explain the sample cohort the very first time in introduction or directly in result section in detail it will be helpful as multiple subgroups are there, it may lead to confusion for the readers.
6. In line 197: "MATN3 promotes expression of HIF-1a" but how it relates with AMS or its symptoms is not lacking, hence look incomplete. If authors feel fit they should add relevance of HIF-1a to AMS.
7. In line 307 to 309: "BILT, 307 CA125, ferritin, FSH, LDL, and prolactin showed significant differences between the nAMS4k and nAMS1k groups but not between the nAMS4k and nAMS1k groups." It seems to have typo as both the groups compared are same.
8. The study is elaborative and well conducted. However, can AMS be a psychological impact as well? Instead of the elevation as such? Have authors tested for a virtual simulation on the control group to observe the effect of feeling to be on heights instead of being on high altitudes physically?
9. In line 332 and 333: p value is given as > 0.05 , kindly confirm if that is right or it should be < 0.05 .
10. In line 371 – "a key rate-limiting enzyme in glycolysis, found in AMS4k than in nAMS4k would lead to the production" found in high or low levels in AMS4K than in nAMS4K?
11. In line 525: Is ADAM15 and inflammation has been observed to exhibit any correlation with any other inflammatory clinical condition? Have authors observed any AMS patient to be prone to other diseases/pathogens when the immune system is activated for AMS condition.
12. In line 575 and 576: "Moreover, the prediction and diagnosis models obtained using both clinical indexes and proteins did not perform as well as those established using only proteins" What may be the potential reason for this? As combination of more perspectives generally give a better segregation, even if not then also proteins would have helped in proper segregation. So, why combination has performed poorly as compared to only proteins?
13. In line 634: "46 paired plasma samples from 22 individuals without AMS" isn't this 23 individuals? Is it a typing error?
14. In lines 630 – 641: The experimental paragraph and sample collection paragraphs can be merged as they are looking iterative.

15. In the section plasma proteome profiling and analysis: what is the reason to select these specific panels only? Any specific reason, as it may lead to biased outcomes?
16. In Line 660: why the undetected features were replaced with 1/10th of the minimum nonzero values? Wont this effect the median or mean of the study for biomarker search?
17. In lines 704 to 707: "Samples with less than 75% of observations were eliminated during the correlation analysis. In addition, features observed in less than 25% of the samples were deleted with the exception of the symptom phenotypes involved in LLS between the AMS4k and AMS1k groups and the AMS4k and nAMS4k groups. Additionally, features with the same value in each sample were deleted." Have authors looked for missing values at random for removing the features as few features may be group specific features, may be a good biomarker.

Reviewer #3 (Remarks to the Author):

Yang et al., reported a study to obtain biomarker for Acute mountain sickness (AMS) through plasma proteomic profilin using a combination of antibody-based proximity extension assay and multiple reaction monitoring (MRM) mass spectrometry along with other clinical features. The authors also included review of previous related study on AMS. Overall, the study is well designed performing rigorous data analysis. The study will provide important insight in AMS and I recommend for publication in Communications Biology with the following revision.

1. The proteomic sample preparation workflow including proteolytic digestion methods need to be described in methodology section.
2. MRM based assay is a gold standard method of biomarker validation with high sensitivity and specificity. However, the performance depends on quality of the developed assay including unique detectable peptide precursor and associated fragments selection and quantification method. The authors briefly mentioned in page 7 line 183 that 102 proteins (538 fragment ions) were monitored. Detailed description of the sequence and number of peptide precursor/fragments per protein will be beneficial. Example of spectra or quantitation result can also be provided to evaluate the signa quality. The transition list can also be provided in supplementary files as a reference for the community who may want to analyze the data using common software like Skyline.

Reviewer #4 (Remarks to the Author):

My comments are here.

1. How many samples overlapped between the discovery and validation phases? Could you show the trend of proteins level of some common proteins identified by PEA and MRM technology?
2. The sample is very limited to the age range (18-20) and male sample. How can you extrapolate the sample result in other ranges and female populations?
3. What is the basis of the selection of fold change criteria of $\text{Log}_2\text{FC} > 0.5$? Please show the power calculation for the same.
4. What are the selection criteria for adding 55 proteins from several pathways to the list of MRM assays involved? Could you provide the list of those proteins?
5. How you have classified the protein list under pathogenesis, predictive, protection, and diagnosis? Please provide any supporting research article for the same, if any.
6. Fig 3B: could you show the trend of proteins from individual samples instead of averaging them?
7. I did not see any reference to proteomic repository data download, or inclusion of proteomic data that fits the criteria for publishing such results. Can you please include the appropriate peptide and protein level data as well as deposit it in an appropriate repository?
8. Line 634: Looks typo error in mentioned line. "60 paired plasma samples from 30 individuals with AMS 633 (including the subjects used for PEA) and 46 paired plasma samples from 22 individuals without AMS 634 were used for validation."
9. Sen Cui et al. 2021, "Novel insights into plasma biomarker candidates in patients with chronic mountain sickness based on proteomics" has considered the 57 samples (30 AMS and 27 control). So how can claim you have used the large cohort. I did not see any big difference in terms of no.

patient's correlation with the existing study.

10. What is the criteria to select the unique peptide and transition for MRM. How have you ensured the quantification of target peptides such as library match or run synthetic peptides along with sample?

11. Could represent the MRM data of potential protein in supp. Figure?

Response to referees:

Below is the point-by-point response to each of those comments.

Reviewer #1

Comments to the Author

Manuscript COMMSBIO-21-3523 entitled “Proteomic and clinical biomarkers for acute mountain sickness diagnosis, prognosis, protection, and pathogenesis in a longitudinal cohort” by Jing Yang, Zhilong Jia, Xinyu Song, Jinlong Shi, Xiaojing Zhao, Kunlun He tries to identify the ascending to high-altitude by non-high-altitude natives is an ideal model for studying acclimatization to extreme environments. However, the Acute mountain sickness (AMS) is frequently experienced by visitors exposed to a high-altitude environment. And, they found and validated that RET played an important role in the pathogenesis of AMS. They prioritized ADAM15, PHGDH, and TRAF2 as protective, predictive, and diagnostic biomarkers, respectively. Moreover, C-peptide, a diabetes biomarker, was actively involved in the pathogenesis, probably assisting with the diagnosis of AMS. The **findings shed light on the precision medicine for AMS and understanding of acclimatization to high-altitude environments**. In my opinion before publication the submitted manuscript needs major revision. You should also correct your paper according to the following points:

We thank the reviewer for the constructive comments and suggestions, which we have fully addressed. Specifically, we made extensive revisions to our initial submission, added additional results to strengthen our conclusions and edited the manuscript extensively.

Comments:

1. When did the subjects evaluate the AMS according to the LLS after ascension to high altitude (4,300 m)?

All subjects completed the questionnaire within 12-24 hours of arrival at high altitude. This has been clarified in the Results section as follows.

"All the participants completed a questionnaire within 12-24 hours of arrival at high altitude, which included 22 AMS symptom phenotypes (Supplementary Data 1), when they were at high altitude prior to blood collection."

2. Twenty paired plasma samples from 10 individuals with AMS at 1 km (AMS1k) and 4 km (AMS4k) were selected. How to identify the protein expression profile using Olink's PEA technology? Why did choice the 10 individuals?

We have provided a detailed description of PEA technology in the Methods section in the revised manuscript. We have also clarified the random selection of 10 individuals in the “PEA-based identification of proteins involved in AMS at the discovery stage” part of the Results section.

1) We have now provided a more thorough introduction to PEA technology.

- PEA technology merges an antibody-based immunoassay, polymerase chain reaction (PCR), and the readout using quantitative real-time PCR (qPCR). Detailed information can be found at the following link: <https://www.olink.com/resources-support/document-download-center/>.
- The panels of PEA involve different diseases and important biological processes, and they can utilize as little as 1 μ L of blood sample in each panel to measure 92 proteins simultaneously. This process results in a scalable, multiplex and highly specific method where the abundance of hundreds of protein biomarkers can be quantified simultaneously.
- In our study, we measured 12 panels (cardiovascular II panel, cardiovascular III panel, inflammation panel, neurology panel, oncology II panel, neuro exploratory panel, development

panel, cardiometabolic panel, immune response panel, cell regulation panel, metabolism panel, and organ damage panel), consisting of 1069 proteins, and we identified 887 proteins. PEA technology is widely used for proteomic profiling and biomarker discovery in many diseases, such as cardiovascular disease.

We have added a detailed description in the Methods section as follows:

“We analyzed 20 plasma samples from individuals with AMS using Olink’s PEA technology with the remaining 12 commercially available panels at that time, excluding 1 panel with overlap in most proteins, by iCarbonX (Shenzhen) Company Limited. These panels consisted of 5 disease panels (Cardiovascular II panel, Cardiovascular III panel, Inflammation panel, Neurology panel, and Oncology II panel) and 7 important biological progress panels (Neuro Exploratory panel, Development panel, Cardiometabolic panel, Immune Response panel, Cell Regulation panel, Metabolism panel, and Organ Damage panel), as indicated on the manufacturer’s website (Olink Proteomics, Uppsala, Sweden). For each panel, each serum sample (1 μ L) was added to 3 μ L of the incubation mix and incubated at 4 $^{\circ}$ C overnight (16-22 h). The extension mix was prepared by mixing PEA enzyme and PCR reagents in nuclease-free water. A total of 96 μ L of extension mix was added to the samples and immediately transferred to the thermal cycler, allowing a 20-min DNA extension at 50 $^{\circ}$ C, followed by 17 cycles of DNA amplification. Further, 2.8 μ L of post-PCR product was mixed with 7.2 μ L of detection mix containing PCR polymerase and real-time PCR reagents in a new 96-well plate. The mixture and PCR primers were loaded onto the primed microfluidic chip (96.96 Dynamic Array IFC, Fluidigm, USA), followed by real-time PCR performed in the Biomark HD system (Fluidigm, USA) using the program provided by Olink Biosciences.”

- 2) Since we measured all 12 panels (>1000 proteins total) available at that time, due to the budget, we randomly selected 10 subjects from 30 AMS patients and used their plasma collected at low altitude and high altitude for proteome profiling. This is still one of the largest proteomics datasets on AMS, with a total of 21,600 measurements (1069 proteins per subject per condition in 10 subjects and 2 conditions and 22 AMS symptom phenotypes in 10 subjects and 1 condition).

In the revised manuscript, we have clarified “PEA-based identification of proteins involved in AMS at the discovery stage” in the Results section.

“At the discovery stage, we measured 21,600 measurements, consisting of 1,069 proteins per subject per condition in 10 subjects (10 AMS) and 2 conditions (1k and 4k), and 22 AMS phenotypes in 10 subjects and 1 condition (4k), to explore the pathogenesis and therapeutic targets of AMS using 12 PEA panels (Supplementary Data 2).”

3. RET played an important role in the pathogenesis of AMS. How to prove the role of in the pathogenesis of AMS?

To strengthen our conclusion, we have added a ROC analysis of RET in the pathogenesis comparison and found an AUC of 0.74 in distinguishing between the AMS4k and AMS1k groups in a new subpanel of **New Supplementary Fig. 9 (previously Supplementary Fig. 4)**.

- 1) Given the interventional nature of our study, where individuals moved from low to high altitude, we have a better ability to infer candidate causal factors compared with simply observational studies. Specifically,

we measured 1069 proteins per subject in 10 individuals with AMS before and after moving to high altitude.

We agree, however, that even this nature-interventional study may have limitations due to other factors, which we describe in the “Limitations and future work” part of the Discussion section as follows:

“Notably, the evidence we provide represents a combination of molecular evidence on the mechanistic roles of these proteins, and the direct observation of biological changes upon intervention in our longitudinal cohort.”

- Moreover, we provide two lines of evidence supporting each candidate. First, we identified candidate pathogenesis-related proteins by comparing the proteome profiles measured via PEA technology between the AMS4k and AMS1k groups. In addition, we validated our initial candidates via MRM technology in an expanded cohort (AMS4k, $n = 30$; AMS1k, $n = 30$, including the 10 paired samples measured by PEA). RET was significantly different between the AMS4k and AMS1k groups in both the discovery and validation stages (**Fig. 3a**).

Fig. 3a Heatmap of the \log_2FC of 75 PEA- or MRM-identified DEPs in the 5 comparison groups based on PEA or MRM. Differential analysis was performed with the identified proteins in 4 comparison groups (pathogenesis: AMS4k and AMS1k; protection: nAMS4k and nAMS1k; prediction: AMS1k and nAMS1k; and diagnosis: AMS4k and nAMS4k). The heatmap depicts the regulation trends of each protein or clinical index with the legend at the left of the figure. The proteins and clinical indexes are labeled with * (q -value < 0.05).

We also performed the XGBoost machine learning model via the differentially expressed proteins between the AMS4k and AMS1k groups. It showed good accuracy (AUC = 0.9) in distinguishing AMS4k and AMS1k with 7 proteins, among which RET had the highest weight (**Fig. 3c**).

Fig. 3c ROCs of the classification models with proteins (orange curve), clinical indexes (blue curve), or combined data (yellow curve) and bar plot of important proteins (orange bar) between AMS4k and AMS1k (pathogenesis, pink box). The p-value of each AUC is shown in parentheses after the AUC value. All the pathogenesis models with three types of features exhibited high accuracy with statistical significance, as demonstrated by the AUCs and p-values.

- 3) We added the classifier performance with RET in distinguishing AMS4k and AMS1k groups (AUC = 0.74, **New Extended Supplementary Fig. 9c, subpanel added in the previous Supplementary Fig. 4**).

Part of Supplementary Fig. 9c. ROC curves of RET in the pathogenesis (pink box) comparison.

- 4) In the correlation analysis between proteins and AMS symptoms, RET was positively correlated with headache ($\rho = 0.35$, p value = 0.06). Notably, headache is the main symptom of AMS and is necessary for the diagnosis of AMS, according to the Lake Louise Acute Mountain Sickness Score (LLS)¹.

Correlation heatmap of proteins and AMS symptom phenotypes in the pathogenesis comparison. The size of the circles corresponds to the absolute value of the correlation coefficients. Significant connections are labeled with *. The correlation between headache and RET is labeled with an arrow.

According to the above evidence, we suggest that RET plays an important role in the pathogenesis of AMS. Notably, the particular underlying mechanism requires further study.

We have clarified the added analysis in the Discussion section as follows:

“RET was identified as the most important variable in the machine learning-based model, with high classifier accuracy (panel, AUC = 0.9; single, AUC = 0.74) for the AMS4k and AMS1k groups (**Fig. 3c**, **Supplementary Fig. 9c**).”

We have also clarified the limitations in the “Limitations and future work” part of the Discussion section as follows:

“Notably, the evidence we provide represents a combination of molecular evidence on the mechanistic roles of these proteins, and the direct observation of biological changes upon intervention in our longitudinal cohort. We plan to further validate these biomarkers using another independent cohort, covering a larger number of individuals, including women, other races, other age ranges, and explore the association between AMS and other phenotypes, such as psychological factors, to contribute to translational medicine.”

4. They prioritized ADAM15, PHGDH, and TRAF2 as protective, predictive, and diagnostic biomarkers, respectively. How to prove ADAM15, PHGDH, and TRAF2 as the protective, predictive, and diagnostic biomarkers?

We confirm that we do not provide “proof” in our biological manuscript.

We have added ROC analyses of ADAM15, PHGDH, and TRAF2 in the protection, prediction, and diagnosis comparison in the revised manuscript to strengthen our conclusion. The ROC curves were added as a subpanel in **Supplementary Fig. 9 (Previously Supplementary Fig. 4)** and showed the classifier performance using ADAM15 in distinguishing the nAMS4k and nAMS1k groups with AUC of 0.80, the classifier performance using PHGDH in distinguishing the AMS1k and nAMS1k groups with AUC of 0.84 and the classifier performance using TRAF2 in distinguishing the AMS4k and nAMS4k groups with AUC of 0.84.

Part of Supplementary Fig. 9c ROC curves of ADAM15, PHGDH, and TRAF2 in protection (dark blue box), prediction (dark green box), and diagnosis (yellow box) comparison, respectively.

We have defined the terms protective, predictive, and diagnostic in our manuscript based on these nature-interventional studies. Specifically,

- We defined “pathogenesis” as a comparison between individuals with AMS at high altitude (4k) and low altitude (1k) in our nature-interventional study. The proteomic signature observed in individuals with AMS at 4k relative to the same individuals at 1k could be used to describe the pathogenesis of AMS.
- We defined “protection” as a comparison between individuals without AMS at 4k and 1k in our nature-interventional study. The proteomic signature observed in individuals without AMS at 4k relative to the same individuals at 1k could be used to describe protection from AMS.
- We defined “prediction” as a comparison between individuals with AMS at 1k and individuals without AMS at 1k in our nature-interventional study. The proteomic signature observed in individuals with AMS vs. individuals without AMS at 1k (before the intervention) could be used to describe the prediction of AMS disease.
- We defined “diagnosis” as a comparison between individuals with AMS at 4k and individuals without AMS at 4k in our nature-interventional study. The proteomic signature observed in individuals with AMS vs. individuals without AMS at 4k (once the change in altitude had already been conducted) could be used to describe the diagnosis of AMS.

We believe these terms are well-justified given the nature-interventional and longitudinal nature of our study, and they will help clarify these comparisons for ease of reading.

We have clarified the classifier performance of each protein in the Discussion section as follows:

“ADAM15 was identified as a candidate protective biomarker of individuals without AMS and was selected by the machine learning-based model, with high accuracy (panel, AUC = 0.95; single, AUC = 0.80) in distinguishing nAMS1k and nAMS4k (Fig. 3d, Supplementary Fig. 9c).”

“PHGDH, which was the top-weighted protein identified by the XGBoost classifier, with an AUC of 0.91 between AMS1k and nAMS1k from the panel and 0.84 from single protein, was identified as a candidate predictive biomarker of AMS (Fig. 3d, Supplementary Fig. 9c).”

“TRAF2 was selected via differential analysis and the XGBoost classifier, with an AUC of 0.89 between AMS4k and nAMS4k, and 0.84 from single protein (Fig. 3f, Supplementary Fig. 9c).”

5. PHGDH is involved in glucose and energy metabolism, and other references were quoted, the results indicate that PHGDH may be a promising predictive biomarker for AMS. TRAF2 is a candidate diagnostic biomarker for AMS, ADAM15 may play a protective role in individuals without AMS. But this is the lack of direct evidence.

We thank the reviewer for the very clear summary of three of the key findings of our paper. We agree that, as with any study conducted in humans outside a clinical trial, it is difficult to establish that a particular protein is indeed driving a response, without directly manipulating that protein in humans, which we hope the reviewers will agree is outside the scope of this manuscript. The evidence we provide is a combination of molecular data on the mechanistic roles of these proteins and the direct observation of biological changes upon intervention in our longitudinal cohort.

We have now clarified these distinctions in the “Limitations and future work” part of the Discussion section in our revised manuscript as follows:

“Notably, the evidence we provide represents a combination of molecular evidence on the mechanistic roles of these proteins, and the direct observation of biological changes upon intervention in our longitudinal cohort. We plan to further validate these biomarkers using another independent cohort, covering a larger number of individuals, including women, other races, other age ranges, and explore the association between AMS and other phenotypes, such as psychological factors, to contribute to translational medicine.”

Reviewer #2

This is an **interesting study** in the direction of searching for the protective, predictive, and diagnostic biomarker as well as for **understanding the pathogenesis**. I would like to **thank the authors** for **such an elaborative and holistic study** that may be **useful for understanding the disease better**. However, there are **small queries and suggestions** that if included, might be helpful for readers.

We thank the reviewer for the very kind comments describing our study as “interesting, elaborative, holistic, and useful” and for the very constructive suggestions that we have fully addressed and that we believe have helped strengthen our manuscript. We provide our point-by-point responses below.

Here are few comments and suggestions:

1. Give full-explanation of abbreviations at the first appearance such as RET, LLS and more.

We thank the reviewer for the suggestion and have updated all the related content.

For example,

“Among these proteins, matrilin 3 (MATN3), myocilin (MYOC), ret proto-oncogene (RET), and S100 calcium binding protein A12 (S100A12) were significantly different (Fig. 3a, Supplementary Fig. 4a,b), which validates their potential significant involvement in the response to AMS.”

“The diagnosis of AMS depends on questionnaire-based diagnostic instruments, among which the Lake Louise Acute Mountain Sickness Score (LLS) is widely used⁴.”

“The levels of anti-inflammatory and/or anti-permeability factors, such as interleukin 1 receptor antagonist (IL-1RA), heat shock protein-70 (HSP-70), and adrenomedullin, are higher in AMS-resistant subjects than in AMS-susceptible subjects, whereas the levels of the chemotactic factors C-C motif chemokine ligand 2 (CCL2) and tumor necrosis factor alpha (TNF- α) are independent of the AMS status¹¹.”

2. In line 40: “providing an ideal model”. Any word like “ideal”, “first time” are big claims and should be avoided.

We thank the reviewer for the comments, which we have incorporated. We have now used “well-suited” (instead of “ideal”) and removed the use of “first time”, and we have revised our manuscript to avoid such statements.

We have clarified this point in the Abstract:

"Ascending to high-altitude by non-high-altitude natives is a well-suited model for studying acclimatization to extreme environments."

We have clarified this point in the Introduction section:

"High-altitude is a typical extreme environment, providing a well-suited model for studying acclimatization."

3. In lines where some facts are stated, an apt reference is required, such as –

We thank the reviewer for the suggestions and have now provided references supporting each of these statements, which we believe has helped improve our manuscript, as follows.

- a. Lines 43- 44 – Definition of AMS with an altitude of above 2500 m

a. Added reference: Luks, A. M., Swenson, E. R. & Bärtsch, P. Acute high-altitude sickness. *Eur Respir Rev* **26**, (2017).²

- b. Line 49 – “multi-factors, such as race, age...” reference for race and age can be mentioned if different from altitude attained reference.

b. Race and age were removed after a systematic literature review.

Meta-analysis conducted in 2018 and 2020 suggested that there is no association between age and the risk of AMS.^{3,4}

Yongjun Luo et al. 2019 reviewed that “there was a statistically significant higher prevalence rate of AMS in women than in men (RR = 1.24, 95%CI 1.09–1.41), regardless of age or race”.⁵

c. Line 197 – “MYOC, is involved in cytoskeletal function”

- c. Added reference: Kubota, R. *et al.* Genomic organization of the human myocilin gene (MYOC) responsible for primary open angle glaucoma (GLC1A). *Biochem Biophys Res Commun* **242**, 396–400 (1998).⁶

d. Line 236 – “CD38 and KITLG are involved in the hematopoietic cell lineage”

- d. Clarified that CD38 and KITLG were involved in the hematopoietic cell lineage according to the KEGG enrichment results (**Supplementary Data 4**).

e. Line 245 and 246 – “Among these DEPs, PCK1 and RBKS are involved in the metabolism of carbohydrates.”

e. Added references

- i. Latorre, P. *et al.* c.A2456C-substitution in Pck1 changes the enzyme kinetic and functional properties modifying fat distribution in pigs. *Sci Rep* **6**, 19617 (2016).⁷
- ii. Kim, Y. *et al.* Function of the pentose phosphate pathway and its key enzyme, transketolase, in the regulation of the meiotic cell cycle in oocytes. *Clinical and experimental reproductive medicine* **39**, 58–67 (2012).⁸
- iii. Marks, P. A. A newer pathway of carbohydrate metabolism; the pentose phosphate pathway. *Diabetes* **5**, 276–283 (1956).⁹

The references above state that pentose phosphate pathway, which involves enzymes such as RBKS, and gluconeogenesis, which involves enzymes such as PCK1 are parts of carbohydrates metabolism.

f. Line 291 – “TRAF2, a TNF receptor-associated factor, is associated with signal transduction from members of the TNF receptor superfamily and apoptosis”

- f. Added reference: Wang, C. Y., Mayo, M. W., Korneluk, R. G., Goeddel, D. V. & Baldwin, A. S. NF- κ B antiapoptosis: induction of TRAF1 and TRAF2 and c-IAP1 and c-IAP2 to suppress caspase-8 activation. *Science* **281**, 1680–1683 (1998).¹⁰

g. In Line 435 – “such as poor appetite, dyspnea, and lip cyanosis”

- g. Clarified that proteins, including FGF23, RET, IL18R1, and GNA14, were positively correlated with AMS symptom phenotypes, such as poor appetite, dyspnea, and lip cyanosis, between the AMS4k and nAMS4k groups, according to **Supplementary Fig. 11b (Originally Supplementary Fig. 6)**.

Supplementary Fig. 11b The network of AMS symptom phenotypes (pink dots), DEPs (blue dots), and clinical indexes (yellow dots) was connected based on the Spearman correlation coefficients and the statistical significance between AMS4k and nAMS4k (diagnosis, yellow box). The edges showing a positive correlation (red line) and a negative correlation (blue line) with q -values less than 0.05 are shown. The thickness of the edges corresponds to the absolute value of the correlation coefficients. The size of the node is proportional to the number of edges from the node. C-peptide showed a positive correlation with proteins that were selected by the XGBoost model or differential analysis and labeled with *.

4. In line 72: authors have mentioned that ANGPTL4 and resistin were associated with AMS, how are they associated? They promoting or helping in suppress the sickness?

Indeed, Barker et al.¹¹ only showed that ANGPTL4 and resistin were increased at high altitude for AMS patients, but they did not provide a detailed mechanistic elucidation of the corresponding pathways, as is typical for such articles, and as these studies can take many additional years.

- 1) Kevin R. Barker et al. recruited 175 subjects to explore biomarkers for acute mountain sickness (AMS) and high-altitude pulmonary edema (HAPE). Of the 175 participants, there were 71 cases of HAPE, 54 cases of AMS, and 50 acclimatized controls (ACs). In comparison to ACs, they found that Angptl4 and

resistin were elevated in AMS ($P = 0.005$; $P = 0.001$, respectively)¹¹.

Figure 1¹¹. Flowchart of study subjects evaluated for biomarkers of altitude illness.

Figure 2¹¹. Box and whisker plots showing the distribution of biomarker levels in climbers returning from altitude. The box represents the median, and interquartile range (IQR), while the whiskers denote the 25 percentile minus 1.5IQR and 75 percentile plus 1.5IQR. Individual data points that fall beyond the whiskers are represented by dots. Data were analyzed using the Kruskal–Wallis test with Dunn’s multiple comparisons test comparing acclimatized healthy controls (AC) vs. AMS, AMS vs. high altitude pulmonary edema (HAPE) and AC vs. HAPE. The stars represent significant comparisons by post-hoc testing (* $P < 0.05$, ** $P < 0.01$, *** $P < 0.001$)

- 2) It is unclear whether ANGPTL4 and resistin promote or help suppress AMS. In Kevin R. Barker et al.’s study, the authors only found an increase in Angptl4 and resistin levels associated with altitude illness. They stated that further study will be required to determine if Angptl4 and resistin play a mechanistic role or are merely reflective of hypoxia¹¹.

We have now clarified this in the Introduction section as follows:

“Kevin et al. found that Angiotensin-like 4 (ANGPTL4) and resistin were increased at a high altitude in AMS patients²⁴.”

5. The sample cohort is a **good number** however, it is a bit **confusing** the way it is mentioned. If authors can be explain the sample cohort the very first time in introduction or directly in result section in detail it will be helpful as multiple subgroups are there, it may lead to confusion for the readers.

We thank the reviewer for stating that our cohort study represents a good number. We also thank the reviewer for the suggestion and have clarified the cohort accordingly.

To clarify this point, we have updated the manuscript with the definitions of all comparisons and the sample cohort in the Results section as follows:

“The discovery stage included two groups from a cohort with 10 individuals with AMS: individuals with AMS at high altitude (AMS4k) and the same individuals at low altitude (AMS1k). The validation stage included four groups from a cohort with 30 individuals with AMS and 23 individuals without AMS: individuals with AMS at high altitude (AMS4k), the same individuals at low altitude (AMS1k), individuals without AMS at high altitude (nAMS4k), and the same individuals at low altitude (nAMS1k).”

“We defined “pathogenesis” as a comparison between individuals with AMS at high-altitude (4k) and low-altitude (1k) in our nature-interventional study. The proteomic signature observed in individuals with AMS at 4k relative to the same individuals at 1k could be used to describe the pathogenesis of the AMS disease. We defined “protection” as a comparison between individuals without AMS at 4k and 1k in our nature-interventional study. The proteomic signature observed in individuals without AMS at 4k relative to the same individuals at 1k could be used to describe protection from AMS disease. We defined “prediction” as a comparison between individuals with AMS at 1k and individuals without AMS at 1k in our nature-interventional study. The proteomic signature observed in individuals with AMS vs. individuals without AMS at 1k (before the intervention) could be used to describe the prediction of AMS disease. We defined “diagnosis” as a comparison between individuals with AMS at 4k and individuals without AMS at 4k in our nature-interventional study. The proteomic signature observed in individuals with AMS vs. individuals without AMS at 4k (once the change in altitude had already been conducted) could be used to describe the diagnosis of AMS disease.”

6. In line 197: “MATN3 promotes expression of HIF-1a” but how it relates with AMS or its symptoms is not lacking, hence look incomplete. If authors feel fit they should add relevance of HIF-1a to AMS.

We thank the reviewer for the suggestion. Acute exposure to hypoxia may lead to AMS. Hypoxia can stimulate hypoxia-inducible factor 1 (HIF-1a) in response¹². Moreover, the HIF-1 signaling pathway is reported to be associated with AMS¹³.

We have clarified the relevance in the “Key proteins involved in the pathogenesis of AMS” part of the Results section as follows:

“MATN3 promotes the expression of HIF-1 α ³¹, the main factor in HIF-1 α pathway, in response to hypoxia and inflammation^{32,33}.”

“In addition, the HIF-1 signaling pathway, glycolysis/gluconeogenesis, and carbon metabolism were enriched in DEPs identified by MRM (Supplementary Fig. 5a and Supplementary Data 4). The HIF-1 signaling pathway is associated with AMS³⁵.”

7. In line 307 to 309: “BILT, 307 CA125, ferritin, FSH, LDL, and prolactin showed significant differences between the nAMS4k and nAMS1k groups but not between the nAMS4k and nAMS1k groups.” It seems to have typo as both the groups compared are same.

We thank the reviewer for catching this typo and apologize for letting that slip into our manuscript.

We have changed the sentence in the “Key clinical indexes of AMS” part of the Results section as follows: “Bilirubin total DPD (BILT), cancer antigen 125 (CA125), ferritin, follicle-stimulating hormone (FSH), low-density lipoprotein (LDL), and prolactin showed significant differences between the nAMS4k and nAMS1k groups but not between the AMS4k and AMS1k groups (Fig. 3b, Supplementary Fig. 7b, Supplementary Data 5).”

8. The study is elaborative and well conducted. However, can AMS be a **psychological impact** as well? Instead of the elevation as such? Have authors tested for a virtual simulation on the control group to observe the effect of feeling to be on heights instead of being on high altitudes physically?

We thank the reviewer for the very interesting hypothesis and possibility. Based on the enriched pathways, the evidence does not suggest that these pathways act in neurons or in the brain and instead seem to indicate their roles in blood, kidney, metabolism, etc. (**Supplementary Fig. 5**). Few studies have focused on this topic.

Samuel J Oliver et al.¹⁴ and Christopher J Boos et al.¹⁵ found a positive correlation between anxiety and AMS. However, we have now noted this very interesting point, searched the related literature, and added it to our Introduction. We may explore this topic in future studies in cooperation with psychologists.

Supplementary Fig. 5 Dot plot of KEGG enrichment of MRM-identified DEPs. **a** KEGG enrichment of MRM-identified DEPs between AMS4k and AMS1k with p-values less than 0.05. Glycolysis/gluconeogenesis, carbon metabolism, and HIF-1 signaling pathways were involved in the development of AMS. **b** KEGG enrichment of the DEPs between the nAMS4k and nAMS1k groups with p-values less than 0.05. The hematopoietic cell lineage was mainly involved in individuals without AMS exposed to high altitude.

We have clarified the related information in the Introduction section as follows:

“The incidence of AMS varies from 25% to 94%, affected by multi-factors, such as anxiety^{5,6}, altitude attained², rate of ascent⁷, individual susceptibility⁸, and preacclimatization⁹.”

We have also clarified this in the “Limitations and future work” part of the Discussion section as follows:

“We plan to further validate these biomarkers using another independent cohort, covering a larger number of individuals, including women, other races, other age ranges, and explore the association between AMS and other phenotypes, such as psychological factors, to contribute to translational medicine.”

9. In line 332 and 333: p value is given as > 0.05, kindly confirm if that is right or it should be <0.05.

We thank the reviewer for the kind reminder. There is no typo here. The p values are presented in parentheses after the AUC in the ROC plot (**Fig. 3e and Fig. 3f**). The p values of AUC predicted by clinical indexes were equal to 0.1041 in the prediction comparison and equal to 0.0656 in the diagnosis comparison. The statistical test of the classifier performance (AUCs < 0.7) using clinical indexes in the prediction and diagnosis comparisons is insignificant (p values more than 0.05).

Fig. 3e ROCs of the classification models with proteins (orange curve), clinical indexes (blue curve), or combined data (yellow curve) and bar plot of important proteins (orange bar) between the AMS1k and nAMS1k groups (prediction, dark green box). The prediction models with proteins and combined data show high accuracy. The p-value of each AUC is shown in parentheses after the AUC value.

Fig. 3f ROCs of the classification models with proteins (orange curve), clinical indexes (blue curve), or combined data (yellow curve) and bar plot of important proteins (orange bar) between the AMS4k and nAMS4k groups (diagnosis, yellow box). The diagnosis models with proteins and combined data exhibit high accuracy with statistical significance. The p-value of each AUC is shown in parentheses after the AUC value.

We have added a detailed explanation in the figure legend for clarity.

“The p value (in parentheses) of each AUC is shown after the AUC value.”

10. In line 371 – “a key rate-limiting enzyme in glycolysis, found in AMS4k than in nAMS4k would lead to the production” found in high or low levels in AMS4K than in nAMS4K?

According to **Fig. 4** and **New Supplementary Fig. 10a** (previously **Supplementary Fig. 5b**), the PFKM was higher in AMS4k than nAMS4k.

We have clarified this issue in the “Changes in carbohydrate metabolism between individuals with and without AMS” part of the Results section as follows:

“First, PFKM, a key rate-limiting enzyme in glycolysis, was found at a higher level in AMS4k than in nAMS4k (Fig. 4 and Supplementary Fig. 10a), which would lead to the production of more fructose 1,6-bisphosphate and thereby the stimulation of glycolysis.”

Fig. 4 Changes in proteins involved in carbohydrate metabolism in AMS. The log₂FC of proteins in glycolysis (light purple background and arrows), gluconeogenesis (light pink background and arrows), the TCA cycle (light green background and arrows), and glycogen synthesis and degradation (light orange background and arrows) in the following 4 comparison groups are shown in the diamond shape: pathogenesis (path, pink background), protection (prot, dark blue background), prediction (pred, dark green background), and diagnosis (diag, yellow background). Carbohydrate metabolism was dysregulated in individuals after ascension to high altitude.

Part of Supplementary Fig. 10a Violin plot PFKM, which is related to carbohydrate metabolism. The protein is presented in the following order: AMS1k (red violin), AMS4k (blue violin), nAMS1k (light yellow violin) and nAMS4k (light green violin). The proteins are labeled with * (q-value < 0.05) and + (p-value < 0.05).

11. In line 525: Is ADAM15 and inflammation has been observed to exhibit any correlation with any other inflammatory clinical condition? Have authors observed any AMS patient to be prone to other diseases/pathogens when the immune system is activated for AMS condition.

We thank the reviewer for the excellent questions. As reviewed by Laetitia Charrier-Hisamuddin et al.¹⁶, ADAM15 is associated with inflammatory clinical conditions, such as atherosclerosis, rheumatoid arthritis, intestinal inflammation, and inherent angiogenesis. It has been reported that the silencing of ADAM15 can inhibit the expression of proinflammatory cytokines in rheumatoid angiogenesis¹⁷.

In our study, we did not find any AMS patient to be prone to other diseases/pathogens when the immune system is activated under AMS conditions. However, severe AMS may develop into fatal high-altitude cerebral edema or high-altitude pulmonary edema, all of which are associated with edema².

We have clarified the related clinical conditions in the “The downregulation of ADAM15 may play a protective role in individuals without AMS” part of the Discussion section as follows:

“ADAM15 is involved in the response to hypoxia, proteolytic ectodomain processing of cytokines, cell adhesion signaling, and angiogenesis in endothelial cells^{71,72}, and it is associated with atherosclerosis, rheumatoid arthritis, intestinal inflammation, and inherent angiogenesis⁷³. In addition, the silencing of ADAM15 can inhibit the expression of proinflammatory cytokines in rheumatoid angiogenesis⁷².”

12. In line 575 and 576: “Moreover, the prediction and diagnosis models obtained using both clinical indexes and proteins did not perform as well as those established using only proteins” What may be the potential reason for this? As combination of more perspectives generally give a better segregation, even if not then also proteins would have helped in proper segregation. So, why combination has performed poorly as compared to only proteins?

We agree with the reviewer’s intuition that more variables should help improve the performance. However, if these variables are spurious or unrelated to the outcome, the inclusion of additional variables can negatively impact performance. Most importantly, with the use of a control group to estimate the expected

classification power of random variables, the inclusion of additional variables can increase the classification performance of even the randomized control group. Thus, the relative performance of the unshuffled data can decrease in comparison, and thus, the reported overall performance can decrease. The observation that performance does not always increase with additional variables is a testament to the accuracy of our machine learning cross-validation setup.

Indeed, we used a 10-fold cross-validation control setting in our model. Because of the strict setting in which our XGBoost machine learning model was applied, cross-validation and comparison with shuffled datasets ensured that spur.

We have clarified this in the “High AMS classification accuracy” part of the Results section.

“We used 10-fold cross-validation and the DEPs measured by MRM assays to enhance the robustness and performance of the models.”

13. In line 634: “46 paired plasma samples from 22 individuals without AMS” isn’t this 23 individuals? Is it a typing error?

We thank the reviewer for catching this typo and apologize for letting that slip into our manuscript. It should be 23 individuals. We have now changed “22 individuals without AMS” to “23 individuals without AMS” in the revised manuscript.

14. In lines 630 – 641: The experimental paragraph and sample collection paragraphs can be merged as they are looking iterative.

We thank the reviewer for the suggestion. We have merged the two paragraphs in the revised manuscript as “**Experimental setup, sample collection, and biochemical detection**” in the Methods section as follows:

“Experimental setup, sample collection, and biochemical detection

All subjects were transported to an altitude of 4,300 m (4 km) from 1,200 m (1 km) within 4 hours by plane. The AMS symptoms were evaluated after ascension to 4 km. Peripheral venous whole-blood samples of 53 subjects (30 AMS and 23 nAMS) were collected at an altitude of 1 km (AMS1k and nAMS1k) and after arrival at an altitude of 4 km (AMS4k and nAMS4k) for 1-4 days. Blood samples were collected in a semirecumbent position from an anterior elbow vein by conventional venipuncture and placed in an EDTA-coated blood collection tube. Plasma was separated by centrifugation and stored in a 0.5 mL aliquot at -80°C until analysis. All samples from these subjects at both time points were also prepared for biochemical detection. Two milliliters of each plasma sample was used to assay 65 clinical indexes (Supplementary Data 1) using a hematology analyzer (cobas 6000; Roche, USA).

Twenty paired plasma samples from 10 individuals with AMS at 1 km (AMS1k) and 4 km (AMS4k) were selected to identify the protein expression profile using Olink’s PEA technology. Subsequently, 60 paired plasma samples from 30 individuals with AMS (including the subjects used for PEA) and 46 paired plasma samples from 23 individuals without AMS were used for validation.”

15. In the section plasma proteome profiling and analysis: what is the reason to select these specific panels only? Any specific reason, as it may lead to biased outcomes?

- 1) When the project was executed, the Olink company had only 13 types of panels; one of them, which had many duplications of arrays with the other 12 panels, was ruled out due to the cost perspective, so we chose the 12 panels for the experimental program at that time.
- 2) We used these Olink panels for the following reasons:
 - a) AMS is characterized by nonspecific symptoms. As reviewed by David R Murdoch et al. in “High-altitude illness”¹⁸, the main symptoms of AMS are headache, poor appetite, nausea, vomiting, fatigue, dizziness, and sleep disturbance, but not all of them need to be present. It involves different systems and biological processes, such as cardiovascular system and inflammation.
 - b) The 12 panels we used in this study consisted of 5 disease panels (cardiovascular II panel, cardiovascular III panel, inflammation panel, neurology panel, and oncology II panel) and 7 important biological progress panels (neuro exploratory panel, development panel, cardiometabolic panel, immune response panel, cell regulation panel, metabolism panel, and organ damage panel), covering 1,069 proteins.
 - c) Robert E. Gerszten et al. reviewed that the PEA assay offers several advantages, including robust multiplexing, low sample consumption, and sensitivity, and it is currently being developed for thousands of proteins and used for large-scale plasma profiling in cardiovascular disease¹⁹.
 - d) These PEA panels cover 1,069 proteins in the plasma, more than those based on DDA or DIA-MS platforms, due to the high complexity of plasma proteome as shown in Petrera et al. 2021²⁰.

According to the above reasons, we selected all the panels available when we executed the test for the proteome profiling of AMS.

We have clarified the reasons in the Methods section as follows:

“We analyzed 20 plasma samples from individuals with AMS using Olink’s PEA technology with the remaining 12 commercially available panels at that time, excluding 1 panel with overlap in most proteins, by iCarbonX (Shenzhen) Company Limited.”

16. In Line 660: why the undetected features were replaced with 1/10th of the minimum nonzero values? Wont this effect the median or mean of the study for biomarker search?

Replacing the missing value with 1/10th of the minimum nonzero values is one of commonly used methods of missing data imputation, although there are many other methods for replacing the missing values. The main reason is that these proteins with missing value are low-abundance proteins.

For example,

“Missing data were replaced by 1/10th of the minimum value (default value 100).” Quote is from a recent study in *Nature Communications*²¹.

“Prior to multivariate analysis missing value imputation was applied and features with > 50% missing values were removed and remaining missing values replaced by 1/5 of the minimum positive value for each feature.” Quote is from a recent study in *Scientific Reports*²².

In our study, we first removed the features with missing values greater than 25% in each group and then replaced the remaining missing values with the 1/10th of minimum nonzero values to avoid bias. There is some impact, but this treatment facilitates the analysis rather than directly removing these features.

We have added the related reference in the revised manuscript.

“The values of undetected features were replaced with 1/10 of the minimum nonzero value⁸³.”

17. In lines 704 to 707: “Samples with less than 75% of observations were eliminated during the correlation analysis. In addition, features observed in less than 25% of the samples were deleted with the exception of the symptom phenotypes involved in LLS between the AMS4k and AMS1k groups and the AMS4k and nAMS4k groups. Additionally, features with the same value in each sample were deleted.” Have authors looked for missing values at random for removing the features as few features may be group specific features, may be a good biomarker.

We thank the reviewer for the reminder. We have clarified the criteria in the revised manuscript.

- 1) Only 5 clinical indexes were removed with a detection rate less than 75% in 106 samples. In practice, we checked the raw data manually and did not find differences in the detection rates of these 5 features between each pair of groups greater than 50%, so we did not regard these features as group-special features.
- 2) In the next analysis, we eliminated features with observations of less than 75% in each group to avoid missing specific features.
- 3) Due to the varying severity of AMS patients in this study, many symptoms may not occur in some individuals, so we excluded unobserved symptoms in less than 25% of the population to ensure that the relationship between symptoms, protein and clinical indicators could be demonstrated as much as possible.

We have added a detailed explanation in the Methods section as follows:

"Samples with less than 75% of observations were eliminated during the correlation analysis. In addition, proteins and clinical indexes observed in less than 75% of the samples were deleted, while symptom phenotypes observed in less than 25% of the samples were deleted with the exception of the symptom phenotypes involved in LLS between the AMS4k and AMS1k groups as well as the AMS4k and nAMS4k groups. Features with a difference in detection rate greater than 50% between the two groups were retained."

Reviewer 3

Yang et al., reported a study to obtain biomarker for Acute mountain sickness (AMS) through plasma proteomic profilin using a combination of antibody-based proximity extension assay and multiple reaction monitoring (MRM) mass spectrometry along with other clinical features. The authors also included review of previous related study on AMS. Overall, the study is **well designed** performing **rigorous data analysis**. The study will provide **important insight in AMS** and I **recommend for publication** in Communications Biology with the following revision.

We thank the reviewer for the very kind comments describing our study as “well designed, rigorous, and important”, for the positive recommendation of publication, and for the very constructive suggestions, which we have fully addressed. We provide our point-by-point responses below.

1. The proteomic sample preparation workflow including proteolytic digestion methods need to be described in methodology section.

We thank the reviewer for the constructive suggestion. We have clarified the proteomic sample preparation workflow of PEA and MRM in the revised manuscript.

- 1) The proteomic sample preparation for PEA was added to the “plasma proteome profiling and analysis” part of the Methods section as follows:

"For each panel, each serum sample (1 μ L) was added to 3 μ L of the incubation mix and incubated at 4 °C overnight (16-22 h). The extension mix was prepared by mixing PEA enzyme and PCR reagents in nuclease-free water. A total of 96 μ L of extension mix was added to the samples and immediately transferred to the thermal cycler, allowing a 20-min DNA extension at 50 °C, followed by 17 cycles of DNA amplification. Further, 2.8 μ L of post-PCR product was mixed with 7.2 μ L of detection mix containing PCR polymerase and real-time PCR reagents in a new 96-well plate. The mixture and PCR primers were loaded onto the primed microfluidic chip (96.96 Dynamic Array IFC, Fluidigm, USA), followed by real-time PCR performed in the Biomark HD system (Fluidigm, USA) using the program provided by Olink Biosciences. The protein concentrations were finally normalized and transformed using internal and interplate controls to adjust for intra- and inter-run variation⁸². More detailed information can be found in the panel-specific validation data documents (www.olink.com/downloads). The expression levels of proteins were represented as linear Normalized Protein eXpression (NPX), a relative quantification scale in arbitrary units."

- 2) The proteomic sample preparation for MRM validation was added in the “DEP validation by multiple reaction monitoring” part of the Methods section as follows:

"A total of 10 μ L of plasma samples were subjected to reduction. We added 20 mM dithiothreitol (DTT) solution and incubated it at 37 °C for 1 h. Subsequently alkylation was performed with sufficient iodoacetamide (IAM) for 1 h at room temperature in the dark. The sample was diluted 4 times by adding a 25 mM ammonium bicarbonate (ABC) buffer. Then, trypsin (trypsin:protein = 1:50) was added and incubated at 37 °C overnight. The next day, 50 μ L of 0.1% FA was added to terminate the digestion. Finally, the samples were all desalted with a C18 cartridge to remove the high urea, and desalted samples were dried by vacuum centrifugation for MRM analyses."

2. MRM based assay is a gold standard method of biomarker validation with high sensitivity and specificity. However, the performance depends on quality of the developed assay including unique detectable peptide precursor and associated fragments selection and quantification method. The authors briefly mentioned in page 7 line 183 that 102 proteins (538 fragment ions) were monitored. Detailed description of the sequence and number of peptide precursor/fragments per protein will be beneficial. Example of spectra or quantitation result can also be provided to evaluate the signa quality. The transition list can also be provided in supplementary files as a reference for the community who may want to analyze the data using common software like Skyline.

We thank the reviewer for the comments describing the validation method in our study, MRM, as the “gold standard method”, “with high sensitivity and specificity”.

We have added lists of 102 MRM-validated proteins, 538 peptide precursor/fragments, and 550 transitions (including 12 transitions of BSA fragments) in three sheets of **New Supplementary data 2**.

Besides, the extracted ion chromatograms showed that the retention time of the transitions of each peptide were consistent, and similar trends were observed in most of the samples.

We have added the extracted ion chromatograms of RET, ADAM15, and PHGDH in **Supplementary Fig. 2**.

Supplementary Figure 2: The sample extracted ion chromatograms of proteins measured by MRM. The extracted ion chromatograms of RET, ADAM15, and PHGDH measured by MRM.

Reviewer #4

My comments are here.

1. How many samples overlapped between the discovery and validation phases? Could you show the trend of proteins level of some common proteins identified by PEA and MRM technology?

We have clarified that a total of 20 samples (AMS, n = 10; nAMS, n = 10) overlapped between the discovery and validation phases. We have also added a heatmap of 23 common proteins identified by PEA and MRM technology in **Supplementary Fig. 3**.

Supplementary Fig. 3 Abundance heatmap of 23 proteins measured by PEA and MRM in pathogenesis comparison. Proteins measured by PEA are shown with ‘.x’, while by MRM are shown with ‘.y’. The gray boxes in the heatmap indicate that the samples were not measured by PEA.

We have rephrased the related sentence in the “DEP validation by multiple reaction monitoring” part of the Methods section as follows:

“MRM was applied to verify the 47 selected DEPs measured by PEA and 55 other pathway-related or interest proteins at the validation stage (Supplementary Data 2), which consisted of 106 samples (30 individuals with AMS and 23 nAMS at 1 km and 4 km, respectively, involving the 10 paired samples measured by PEA) by Beijing Qinglian Biotech Company Limited.”

We have rephrased the related sentence in the “Key proteins involved in the pathogenesis of AMS” part of the Results section as follows:

“We confirmed that 23 out of 47 proteins (49%) exhibited changing trends that were consistent with those found for the PEA-identified DEPs (Supplementary Data 3, Supplementary Fig. 3).”

2. The sample is very limited to the age range (18-20) and male sample. How can you extrapolate the sample result in other ranges and female populations?

Given our sample size, we did not have the power to study the effects across multiple age groups and biological sexes. Thus, our study design narrowed both the age range and sex to maximize our discovery power by minimizing the number of other contributing variables to the molecular variation.

- 1) Yes, it would be more perfect if we recruited individuals from all age ranges and included females. The recruitment of volunteers for blood samples on the plain and plateau is not easy. Those willing to participate in this study were mainly male, who also represented the main population for construction and tourism.
- 2) Meta-analysis conducted in 2018 and 2020 suggested that there is no association between age and the risk of AMS^{3,4}. A meta-analysis performed in 2019 suggested that females are more likely to suffer from AMS than males⁵.

According to your suggestion, we have added this limitation to the updated manuscript as follows:

The “Limitations and future work” section of the Discussion section is as follows:

“We plan to further validate these biomarkers using another independent cohort, covering a larger number of individuals, including women, other races, other age ranges, and explore the association between AMS and other phenotypes, such as psychological factors, to contribute to translational medicine.”

3. What is the basis of the selection of fold change criteria of Log2FC >0.5? Please show the power calculation for the same.

To clarify, we did not solely use a fold-change threshold but the combination of fold-change and p value. We calculated the power using the pwr package in R with effect size of 0.8. The power of 10 paired samples used in the PEA discovery stage was 0.62. The power of the test in the pathogenesis comparison consisted of 30 paired samples measured by MRM and was 0.99. The power of 23 paired samples from the protective comparison measured by MRM was 0.96. The power of unpaired samples from the predictive/diagnosis comparison measured by MRM was 0.81.

We could have simply required a q-value threshold, but we imposed the additional log₂FC threshold to ensure that not only significant but also noticeable changes were studied.

An absolute value of log₂FC > 0.5 was used as an intermediate filtering condition to select the candidate proteins. This criterion is arbitrary. The cutoff of 0.5 is also widely used in other literature.

For example, ErwinM Schoof et al. stated that the "Absolute log₂FC difference of proteins of LSC vs. blast between scMS and MS3 bulk-sorted data across intensity bins. Only proteins that were significantly changed between LSC and blast in the MS3 bulk-sorted data were selected for comparison (FDR < 0.05, absolute log₂FC > 0.5)."²³

4. What are the selection criteria for adding 55 proteins from several pathways to the list of MRM assays involved? Could you provide the list of those proteins?

We selected proteins related to carbohydrate metabolism, such as glycolysis/gluconeogenesis (G6PC, ALDOA, PDHA1, PCK1), TCA cycle (PDHA1), proteins associated with the PEA-identified DEPs obtained from the literature (such as SLC4A1, HBB, and SLC2A1), and proteins of interest from previous studies.

We have clarified the selection criteria in **Supplementary Data 2**.

- 1) The 55 proteins were selected based on the following criteria:
 - a) CARHSP1 had more than 70% missing values in one group but no missing values in another group. To confirm their expression trend, we validated them with MRM in the validation cohort.
 - b) According to the KEGG enrichment results of PEA-identified proteins, we found carbon metabolism-related pathways, such as the pentose phosphate pathway and glycolysis (**Fig 2c**). To check the carbon metabolism changes in subjects with/without AMS after arrival at high altitude, we selected some key enzymes in glycolysis/gluconeogenesis and the TCA cycle (PDHA1, PCK1, G6PC, LDHA, PGM1, SDHA) to further explore the changes in different subjects at different altitudes. Since an in-depth study of metabolic changes also requires a combination with metabolomics and is not the focus of this study, only some key enzyme assays were selected.

Fig. 2c Abundance heatmap and KEGG pathway enrichment diagram of the PEA-identified DEPs. The scaled abundance heatmap of DEPs between the AMS4k and AMS1k groups per sample is shown on the left side. The hierarchical cluster analysis in the heatmap showed that the 47 DEPs could clearly distinguish the AMS4k group (red hierarchical tree) and AMS1k group (blue hierarchical tree), and the DEPs could be separated into cluster 1 (purple hierarchical tree) and cluster 2 (brown hierarchical tree). Dot plots of the KEGG enrichment of the two clusters with p-values less than 0.05 are presented on the right side. Cytokine–cytokine receptor interactions and the TNF signaling pathway were enriched in cluster 1 (purple fonts), and nitrogen metabolism and the pentose phosphate pathway were mainly enriched in cluster 2 (brown fonts).

c) The remaining proteins were selected based on proteins associated with PEA-identified DEPs obtained from the literature (such as SLC4A1, HBB, and SLC2A1) or proteins of interest from previous studies.

2) The 55 proteins are listed below (**Supplementary Data 2**) and we clarified this point in the Methods section as follows:

"MRM was applied to verify the 47 selected DEPs measured by PEA and 55 other pathway-related or interest proteins at the validation stage (Supplementary Data 2)"

Source	Symbol	Full Name
Carbon metabolism-related proteins	PDHA1	pyruvate dehydrogenase
Carbon metabolism-related proteins	LDHA	lactate dehydrogenase A

Carbon metabolism-related proteins	PHGDH	phosphoglycerate dehydrogenase
Carbon metabolism-related proteins	PGM1	phosphoglucomutase 1
Carbon metabolism-related proteins	ALDOA	aldolase, fructose-bisphosphate A
Carbon metabolism-related proteins	PCK1	phosphoenolpyruvate carboxykinase 1
Other proteins of interest	PLA2G7	phospholipase A2 group VII
Other proteins of interest	SEMG2	semenogelin II
Other proteins of interest	DDAH2	dimethylarginine dimethylaminohydrolase 2
Other proteins of interest	FLOT2	flotillin 2
Other proteins of interest	TRAF2	TNF receptor associated factor 2
Other proteins of interest	FN1	fibronectin 1
Other proteins of interest	HLA-C	major histocompatibility complex, class I, C
Other proteins of interest	CFL1	cofilin 1
Other proteins of interest	FOS	Fos proto-oncogene, AP-1 transcription factor subunit
Other proteins of interest	B2M	beta-2-microglobulin
Other proteins of interest	CAPZB	capping actin protein of muscle Z-line beta subunit
Other proteins of interest	APCS	amyloid P component, serum
Other proteins of interest	AKAP3	A-kinase anchoring protein 3
Other proteins of interest	POR	cytochrome p450 oxidoreductase
Other proteins of interest	GC	GC, vitamin D binding protein
Other proteins of interest	ITGAV	integrin subunit alpha V
Other proteins of interest	PICALM	phosphatidylinositol binding clathrin assembly protein
Other proteins of interest	ACTR2	ARP2 actin related protein 2 homolog
Other proteins of interest	SLC2A1	solute carrier family 2 member 1

Other proteins of interest	SLC4A1	solute carrier family 4 member 1
Other proteins of interest	EHD3	EH domain containing 3
Other proteins of interest	ARF6	ADP ribosylation factor 6
Other proteins of interest	S100A6	S100 calcium binding protein A6
Other proteins of interest	PHKA2	phosphorylase kinase regulatory subunit alpha 2
Other proteins of interest	ACE2	angiotensin I converting enzyme 2
Other proteins of interest	AKAP4	A-kinase anchoring protein 4
Other proteins of interest	UBA1	ubiquitin like modifier activating enzyme 1
Other proteins of interest	NDRG1	N-myc downstream regulated 1
Other proteins of interest	IGFBP2	insulin like growth factor binding protein 2
Other proteins of interest	GNA14	G protein subunit alpha 14
Other proteins of interest	THY1	Thy-1 cell surface antigen
Other proteins of interest	CD14	CD14 molecule
Other proteins of interest	IGFBP7	insulin like growth factor binding protein 7
Other proteins of interest	EBI3	Epstein–Barr virus induced 3
Other proteins of interest	SPHK1	sphingosine kinase 1
Other proteins of interest	TKT	transketolase
Other proteins of interest	GNA13	G protein subunit alpha 13
Other proteins of interest	AGT	angiotensinogen
Other proteins of interest	PRKACB	protein kinase cAMP-activated catalytic subunit beta
Other proteins of interest	AKT1	AKT serine/threonine kinase 1
Other proteins of interest	PLAU	plasminogen activator, urokinase
Other proteins of interest	PLD1	phospholipase D1
Other proteins of interest	HBB	hemoglobin subunit beta
Other proteins of interest	MAP2K1	mitogen-activated protein kinase

		kinase 1
Other proteins of interest	TNR	tenascin R
Other proteins of interest	G6PC	glucose-6-phosphatase catalytic subunit
Other proteins of interest	MUC6	mucin 6, oligomeric mucus/gel-forming
Other proteins of interest	SDHA	succinate dehydrogenase complex flavoprotein subunit A

5. How you have classified the protein list under pathogenesis, predictive, protection, and diagnosis? Please provide any supporting research article for the same, if any.

We thank the reviewer for the constructive suggestions. We have clarified the definition in the revised manuscript.

We have defined the terms protective, predictive, and diagnostic in our manuscript based on these nature-interventional studies. Specifically,

- We defined “pathogenesis” as a comparison between individuals with AMS at high altitude (4k) and low altitude (1k) in our nature-interventional study. The proteomic signature observed in individuals with AMS at 4k relative to the same individuals at 1k could be used to describe the pathogenesis of AMS.
- We defined “protection” as a comparison between individuals without AMS at 4k and 1k in our nature-interventional study. The proteomic signature observed in individuals without AMS at 4k relative to the same individuals at 1k could be used to describe protection from AMS disease.
- We defined “prediction” as a comparison between individuals with AMS at 1k and individuals without AMS at 1k in our nature-interventional study. The proteomic signature observed in individuals with AMS vs. individuals without AMS at 1k (before the intervention) could be used to describe the prediction of AMS disease.
- We defined “diagnosis” as a comparison between individuals with AMS at 4k and individuals without AMS at 4k in our nature-interventional study. The proteomic signature observed in individuals with AMS vs. individuals without AMS at 4k (once the change in altitude had already been conducted) could be used to describe the diagnosis of AMS.

We believe these terms are well-justified given the nature-interventional and longitudinal nature of our study, and that they could help to clarify these comparisons for ease of reading.

For example,

Hui Lu et al. compared plasma samples from AMS-susceptible individuals at BL status (AMS + BL) with those at 9 h after exposure to high altitude (AMS + 9) to find a potential correlation with the human response to hypobaric hypoxia and AMS.²⁴

Yuqi Gao et al. compared the microRNA array screening results from AMS subjects with those from non-AMS subjects at low altitude to select the predictor of AMS.²⁵

Chengbin Wang et al. compared the acute phase proteins and inflammatory cytokines from AMS subjects with those from non-AMS subjects at high altitude for the diagnosis of AMS.²⁶

We have added the definition in the Results section as follows:

“We defined “pathogenesis” as a comparison between individuals with AMS at high-altitude (4k) and low-altitude (1k) in our nature-interventional study. The proteomic signature observed in individuals with AMS at 4k relative to the same individuals at 1k could be used to describe the pathogenesis of the AMS disease. We defined “protection” as a comparison between individuals without AMS at 4k and 1k in our nature-interventional study. The proteomic signature observed in individuals without AMS at 4k relative to the same individuals at 1k could be used to describe protection from AMS disease. We defined “prediction” as a comparison between individuals with AMS at 1k and individuals without AMS at 1k in our nature-interventional study. The proteomic signature observed in individuals with AMS vs. individuals without AMS at 1k (before the intervention) could be used to describe the prediction of AMS disease. We defined “diagnosis” as a comparison between individuals with AMS at 4k and individuals without AMS at 4k in our nature-interventional study. The proteomic signature observed in individuals with AMS vs. individuals without AMS at 4k (once the change in altitude had already been conducted) could be used to describe the diagnosis of AMS disease.”

6. Fig 3B: could you show the trend of proteins from individual samples instead of averaging them?

We have added three supplementary figures (**Supplementary Fig. 4, 6-7**) to show the trend of differentially expressed proteins and clinical indexes presented in **Fig. 3a** and **Fig. 3b** from individual samples. All p-values were adjusted by Bonferroni-Hochberg corrections.

We also updated **Fig. 3b** with a correction of \log_2FC of the pathogenesis and protection cs in the heatmap. When plotting the heat map, the split function with automatically sorted function in the code led to the calculation of FC with AMS1k divided by AMS4k and nAMS1k divided by nAMS4k in the pathogenesis and the protection comparison group, respectively. Therefore, we got opposite regulation trends in the pathogenesis and the protection comparison group.. The violin plots were made by the values of each sample specifically.

To intuitively visualize the significance of the differential expression of proteins and clinical indexes in different comparison groups as well as the expression trends, we used the union set with four groups of significantly different proteins or clinical indexes, \log_2FC values, and asterisks/plus signs for plotting (**New Fig. 3a, b**).

New Fig. 3a,b Differential analysis with PEA-/MRM-identified DEPs and clinical indexes. Differential analysis was performed with the identified proteins and clinical indexes in 4 comparison groups (pathogenesis: AMS4k and AMS1k; protection: nAMS4k and nAMS1k; prediction: AMS1k and nAMS1k; and diagnosis: AMS4k and nAMS4k). **a** Heatmap of the log₂FC of 75 PEA- or MRM-identified DEPs in the 5 comparison groups based on PEA or MRM. **b** Heatmap of the log₂FC of 37 differential clinical indexes involved in the 4 comparison groups. The heatmap depicts the regulation trends of each protein or clinical index with the legend at the left of the figure. The proteins and clinical indexes are labeled with * (q-value < 0.05) and + (p-value < 0.05).

As suggested by the reviewer, we have generated heatmaps of differential proteins and differential clinical indexes between the different comparison groups (AMS4k vs. AMS1k, nAMS4k vs. nAMS1k, AMS1k vs. nAMS1k, AMS4k vs. nAMS4k) to show the trends from individual samples.

- 1) We developed heatmaps using the differentially expressed proteins (q-values < 0.05) of each comparison.
 - a) We generated two heatmaps of differentially expressed proteins (q-values < 0.05) in pathogenesis comparison due to MRM and PEA results.

A total of 22 DEPs were identified by MRM in the pathogenesis group (**new Extended Supplementary Fig. 4a**). In our study, we focused on MRM-validated DEPs with the same regulatory trend as PEA-identified DEPs for the verification of pathogenesis-related proteins. Filtering by the criteria above, we finally plotted a heatmap of 4 key DEPs (**new Extended Supplementary Fig. 4b**). Comparing the AMS4k group with the AMS1k group, RET and MATN3 were upregulated in most of the samples, while S100A12 and MYOC were downregulated in most of the samples.

Supplementary Fig. 4a, b Heatmap of differentially expressed proteins in pathogenesis comparison.
a Heatmap of 22 differentially expressed proteins (q-values < 0.05) in pathogenesis comparison validated by MRM. **b** Heatmap of 4 differentially expressed proteins (q-values < 0.05) with the same regulation trends measured by both PEA and MRM in the pathogenesis comparison.

- b) We created two heatmaps of differentially expressed proteins (q-values < 0.05) in the protection comparison due to MRM and PEA results.

A total of 29 DEPs were identified by MRM in the protection group (**new extended Supplementary Fig. 4c**). In our study, we focused on MRM-validated DEPs with the opposite regulation trend as PEA-identified and MRM-validated DEPs in the pathogenesis comparison for the discovery of protective proteins. Filtering by the criteria above, we finally made a heatmap of 5 key DEPs (**New extended Supplementary Fig. 4d**). Comparing the nAMS4k group with the nAMS1k group, ADAM15, CST6, KITLG, CD38, and THBD were downregulated in most of the samples.

Supplementary Fig. 4c,d Heatmap of differentially expressed proteins in protection comparison. c Heatmap of 29 differentially expressed proteins (q-values < 0.05) in protection comparison validated by MRM. **d** Heatmap of 5 differentially expressed proteins (q-values < 0.05) with the opposite regulation trends measured by PEA and MRM in the protection comparison.

- c) We created a heatmap of differentially expressed proteins (q-values < 0.05) in the prediction comparison.

A total of 23 DEPs were identified by MRM in the prediction group (**new extended Supplementary Fig. 6a**). Comparing the AMS1k group with the nAMS1k group, proteins such as PHGDH and RBKS were completely different.

Supplementary Fig. 6a Heatmap of differentially expressed proteins in prediction comparison. a
Heatmap of 23 differentially expressed proteins (q-values < 0.05) in prediction comparison validated by MRM.

- d) We have created a heatmap of differentially expressed proteins (q-values < 0.05) in the diagnosis comparison.

A total of 28 DEPs were identified by MRM in the diagnosis comparison (**New Extended Supplementary Fig. 6b**). Proteins such as TRAF2 and MYOC were differentially expressed between AMS4k and nAMS4k groups.

Supplementary Fig. 6b Heatmap of differentially expressed proteins in diagnosis comparison. b Heatmap of 28 differentially expressed proteins (q -values < 0.05) in prediction comparison validated by MRM.

2) We created heatmaps using the differentially expressed clinical indexes of each comparison.

a) We generated a heatmap of 26 differentially expressed clinical indexes (q -values < 0.05) in the pathogenesis comparison (**New Extended Supplementary Fig. 7a**).

Supplementary Fig. 7a Heatmap of 26 differentially expressed clinical indexes (q -values < 0.05) in

pathogenesis comparison.

- b) We generated a heatmap of 24 differentially expressed clinical indexes (q-values < 0.05) in the protection comparison (**New Extended Supplementary Fig. 7b**).

Supplementary Fig. 7b Heatmap of 24 differentially expressed clinical indexes (q-values < 0.05) in protection comparison.

- c) We created a heatmap of 4 differentially expressed clinical indexes (p-values < 0.05) in the prediction comparison (**New Extended Supplementary Fig. 7c**). Since no clinical indexes showed q-value less than 0.05, we used clinical indexes with p-values less than 0.05 for plotting.

Supplementary Fig. 7c Heatmap of 4 differentially expressed clinical indexes (p-values < 0.05) in prediction comparison.

- d) Heatmap of 6 differentially expressed clinical indexes (p-values < 0.05) in diagnosis comparison (**New Extended Supplementary Fig. 7d**). Since no clinical indexes showed q-value less than 0.05, we used clinical indexes with p-values less than 0.05 for plotting.

Supplementary Fig. 7d Heatmap of 6 differentially expressed clinical indexes (p-values < 0.05) in diagnosis comparison.

7. I did not see any reference to proteomic repository data download, or inclusion of proteomic data that fits the criteria for publishing such results. Can you please include the appropriate peptide and protein level data as well as deposit it in an appropriate repository?

We thank the reviewer for the reminder. The related URL link in the availability of data section was broken. We have updated this information in the revised manuscript in the Availability of data section. It will be public after the paper is published.

1) We have updated the proteomic repository data in the “Availability of data” section as follows:

“The raw MRM proteomic data analyzed in this study are available at iProX⁸⁹ with the corresponding dataset identifier [PXD029063](https://www.iprox.cn/page/PSV023.html?url=1643081969878VyeL) (https://www.iprox.cn/page/PSV023.html?url=1643081969878VyeL, temporary password: 3gUW).”

2) We have also placed the peptide and protein levels in the proteomic repository data at the author’s GitHub (https://github.com/Monica1227/AMS_biomarker/tree/main/data)

8. Line 634: Looks typo error in mentioned line. “60 paired plasma samples from 30 individuals with AMS 633 (including the subjects used for PEA) and 46 paired plasma samples from 22 individuals without AMS 634 were used for validation.”

We thank the reviewer for catching this typo and apologize for letting it slip into our manuscript. We have changed this text to “Sixty paired plasma samples from 30 individuals with AMS 633 (including the subjects used for PEA) and 46 paired plasma samples from **23** individuals without AMS” in the revised manuscript.

9. Sen Cui et al. 2021, “Novel insights into plasma biomarker candidates in patients with chronic mountain sickness based on proteomics” considered 57 samples (30 AMS and 27 control). So how can claim you have used the **large cohort**. I did not see any big difference in terms of no. Patient’s correlation with the existing study.

We thank the reviewer for the comment. Sen Cui et al. focused on chronic mountain sickness, not acute mountain sickness. Moreover, we have clarified that since our samples were paired, we had a total of 106 samples, which is approximately double. Still, the article of Sen Cui et al. is an interesting study related to high-altitude.

We have clarified this difference in the Introduction section as follows:

“People traveling to high-altitude regions have an increased risk of developing high-altitude illness, such as acute mountain sickness (AMS)¹, high-altitude pulmonary edema (HAPE)², and chronic mountain

sickness (CMS)³.”

We have also clarified the measurements in our cohort in the Discussion section as follows:

“We measured a total of 40,248 measurements including 21,380 measurements by PEA, 10,812 measurements by MRM, 6,890 measurements of clinical indexes, and 1,166 measurements of AMS symptom phenotypes.”

10. What is the criteria to select the unique peptide and transition for MRM. How have you ensured the quantification of target peptides such as library match or run synthetic peptides along with sample?

We thank the reviewer for the professional questions. We used data-dependent acquisition (DDA) technology for unique peptide selection. For those proteins not identified by DDA, we performed further screening with the transition library in SRMatlas (<https://db.systemsbiology.net/sbeams/cgi/PeptideAtlas/GetTransitions>). Three transitions were selected for each peptide, and two peptides were kept for each protein. We did not generate synthetic peptides in this study, which may be very costly and time-consuming; additionally, it was beyond the scope of our manuscript. We quantified the peptide segment based on the match with the Uniprot Homo sapiens proteome database (canonical including isoforms) downloaded April 2019 (73,645 entries).

In addition, the extracted ion chromatograms showed that the retention time of the transitions of each peptide were consistent, and similar trends were observed in most of the samples.

We have clarified the criteria for selecting the unique peptide and transition for MRM in the Methods section as follows:

“The unique peptide and transition for MRM from the peptides identified by TripleTOF 5600+ mass spectrometry (SCIEX) in the mixed plasma samples were selected using the Skyline software from a background database of human species, and further screened with the transition library in SRMatlas (<https://db.systemsbiology.net/sbeams/cgi/PeptideAtlas/GetTransitions>). Three transitions were selected for each peptide, and two peptides were kept for each protein.”

We have also added the extracted ion chromatograms of RET, ADAM15, PHGDH, and TRAF2 in **Supplementary Fig. 2**.

Supplementary Figure 2: The sample extracted ion chromatograms of proteins measured by MRM. The extracted ion chromatograms of RET, ADAM15, and PHGDH measured by MRM.

11. Could represent the MRM data of potential protein in supp. Figure?

We have uploaded the quantified MRM data at the author's GitHub (https://github.com/Monica1227/AMS_biomarker/tree/main/data).

We have also added the extracted ion chromatograms of RET, ADAM15, PHGDH, and TRAF2 in **Supplementary Fig. 2**.

RET_YTSTLLPGDTWAQQTFR_b6

Serum_MRM_33_PB060-52_1_50min_2...01022.wiff), (sample Index: 1)
Area: 3.335e4, Height: 4.147e3, RT: 25.73 min

RET_YTSTLLPGDTWAQQTFR_b10

Serum_MRM_33_PB060-52_1_50min_2...01022.wiff), (sample Index: 1)
Area: 8.993e4, Height: 9.084e3, RT: 25.78 min

RET_YTSTLLPGDTWAQQTFR_y8

Serum_MRM_33_PB060-52_1_50min_2...01022.wiff), (sample Index: 1)
Area: 1.876e5, Height: 1.885e4, RT: 25.74 min

ADAM15_EPLEPQVLQDDLPISLK_b6

Serum_MRM_49_PB003-14_2_50min_2...01022.wiff), (sample Index: 2)
Area: 7.760e4, Height: 9.549e3, RT: 19.32 min

ADAM15_EPLEPQVLQDDLPISLK_y6

Serum_MRM_49_PB003-14_2_50min_2...01022.wiff), (sample Index: 2)
Area: 4.165e5, Height: 6.055e4, RT: 19.31 min

ADAM15_EPLEPQVLQDDLPISLK_y7

Serum_MRM_49_PB003-14_2_50min_2...01022.wiff), (sample Index: 2)
Area: 2.503e5, Height: 3.191e4, RT: 19.30 min

PHDGH(SERA)- DLPLLLFR-b5

Serum_MRM_28_PB054-79_2_50min_2...01022.wiff), (sample Index: 2)
Area: 7.641e4, Height: 1.333e4, RT: 21.42 min

PHDGH(SERA)- DLPLLLFR-y5

Serum_MRM_28_PB054-79_2_50min_2...01022.wiff), (sample Index: 2)
Area: 3.007e3, Height: 9.974e2, RT: 21.41 min

PHDGH(SERA)- DLPLLLFR-y6

Serum_MRM_28_PB054-79_2_50min_2...01022.wiff), (sample Index: 2)
Area: 5.535e4, Height: 1.036e4, RT: 21.42 min

Supplementary Figure 2: The sample extracted ion chromatograms of proteins measured by MRM. The extracted ion chromatograms of RET, ADAM15, and PHGDH measured by MRM.

Reference:

1. Roach, R. C. *et al.* The 2018 Lake Louise Acute Mountain Sickness Score. *High Alt. Med. Biol.* **19**, 4–6 (2018).
2. Luks, A. M., Swenson, E. R. & Bärtsch, P. Acute high-altitude sickness. *Eur. Respir. Rev.* **26**, (2017).
3. Wu, Y., Zhang, C., Chen, Y. & Luo, Y.-J. Association between acute mountain sickness (AMS) and age: a meta-analysis. *Mil Med Res* **5**, 14 (2018).
4. Gianfredi, V., Albano, L., Basnyat, B. & Ferrara, P. Does age have an impact on acute mountain sickness? A systematic review. *J. Travel Med.* **27**, (2020).
5. Hou, Y.-P. *et al.* Sex-based differences in the prevalence of acute mountain sickness: a meta-analysis. *Mil Med Res* **6**, 38 (2019).
6. Kubota, R. *et al.* Genomic organization of the human myocilin gene (MYOC) responsible for primary open angle glaucoma (GLC1A). *Biochem. Biophys. Res. Commun.* **242**, 396–400 (1998).
7. Latorre, P. *et al.* c.A2456C-substitution in Pck1 changes the enzyme kinetic and functional properties modifying fat distribution in pigs. *Sci. Rep.* **6**, 19617 (2016).
8. Kim, Y. *et al.* Function of the pentose phosphate pathway and its key enzyme, transketolase, in the regulation of the meiotic cell cycle in oocytes. *Clin. Exp. Reprod. Med.* **39**, 58–67 (2012).
9. Marks, P. A. A newer pathway of carbohydrate metabolism; the pentose phosphate pathway. *Diabetes* **5**, 276–283 (1956).
10. Wang, C. Y., Mayo, M. W., Korneluk, R. G., Goeddel, D. V. & Baldwin, A. S., Jr. NF-kappaB antiapoptosis: induction of TRAF1 and TRAF2 and c-IAP1 and c-IAP2 to suppress caspase-8 activation. *Science* **281**, 1680–1683 (1998).
11. Barker, K. R. *et al.* Biomarkers of hypoxia, endothelial and circulatory dysfunction among climbers in Nepal with AMS and HAPE: a prospective case-control study. *J. Travel Med.* **23**, (2016).
12. Balamurugan, K. HIF-1 at the crossroads of hypoxia, inflammation, and cancer. *Int. J. Cancer* **138**, 1058–

- 1066 (2016).
13. Yan, X. *et al.* Salidroside orchestrates metabolic reprogramming by regulating the Hif-1 α signalling pathway in acute mountain sickness. *Pharm. Biol.* **59**, 1540–1550 (2021).
 14. Oliver, S. J. *et al.* Physiological and psychological illness symptoms at high altitude and their relationship with acute mountain sickness: a prospective cohort study. *J. Travel Med.* **19**, 210–219 (2012).
 15. Boos, C. J. *et al.* The relationship between anxiety and acute mountain sickness. *PLoS One* **13**, e0197147 (2018).
 16. Charrier-Hisamuddin, L., Laboisse, C. L. & Merlin, D. ADAM-15: a metalloprotease that mediates inflammation. *FASEB J.* **22**, 641–653 (2008).
 17. Nishimi, S., Isozaki, T., Wakabayashi, K., Takeuchi, H. & Kasama, T. A Disintegrin and Metalloprotease 15 is Expressed on Rheumatoid Arthritis Synovial Tissue Endothelial Cells and may Mediate Angiogenesis. *Cells* **8**, (2019).
 18. Basnyat, B. & Murdoch, D. R. High-altitude illness. *Lancet* **361**, 1967–1974 (2003).
 19. Smith, J. G. & Gerszten, R. E. Emerging Affinity-Based Proteomic Technologies for Large-Scale Plasma Profiling in Cardiovascular Disease. *Circulation* **135**, 1651–1664 (2017).
 20. Petrera, A. *et al.* Multiplatform Approach for Plasma Proteomics: Complementarity of Olink Proximity Extension Assay Technology to Mass Spectrometry-Based Protein Profiling. *J. Proteome Res.* **20**, 751–762 (2021).
 21. Ding, J. *et al.* A metabolome atlas of the aging mouse brain. *Nat. Commun.* **12**, 6021 (2021).
 22. Reddy, P. *et al.* Effects of ergotamine on the central nervous system using untargeted metabolomics analysis in a mouse model. *Sci. Rep.* **11**, 19542 (2021).
 23. Schoof, E. M. *et al.* Quantitative single-cell proteomics as a tool to characterize cellular hierarchies. *Nat. Commun.* **12**, 3341 (2021).
 24. Lu, H. *et al.* Plasma proteomic study of acute mountain sickness susceptible and resistant individuals. *Sci. Rep.* **8**, 1265 (2018).
 25. Liu, B. *et al.* A Signature of Circulating microRNAs Predicts the Susceptibility of Acute Mountain Sickness. *Front. Physiol.* **8**, 55 (2017).
 26. Wang, C. *et al.* Exploration of Acute Phase Proteins and Inflammatory Cytokines in Early Stage Diagnosis

of Acute Mountain Sickness. *High Alt. Med. Biol.* **19**, 170–177 (2018).

Reviewers' comments:

Reviewer #1 (Remarks to the Author):

Author identified candidate pathogenesis-related proteins by comparing the proteome profiles measured via PEA technology between the AMS4k and AMS1k groups. In addition, they validated their initial candidates via MRM technology in an expanded cohort (AMS4k, n = 30; AMS1k, n = 30, including the 10 paired samples measured by PEA). Could you tell the incidence of AMS?

Reviewer #2 (Remarks to the Author):

The authors have done a wonderful job!

I have no more comments. Thank you for taking out time and addressing all the queries raised.

Reviewer #3 (Remarks to the Author):

The authors has provided well revised version of the manuscript and my concerns were all addressed.

Reviewer #4 (Remarks to the Author):

Most of the comments have been addressed. I need more clarification related to my 1st comment.

1. How many samples overlapped between the discovery and validation phases? Could you show the trend of proteins level of some common proteins identified by PEA and MRM technology? We have clarified that a total of 20 samples (AMS, n = 10; nAMS, n = 10) overlapped between the discovery and validation phases. We have also added a heatmap of 23 common proteins identified by PEA and MRM technology in Supplementary Fig. 3.

Comment: Most of the proteins did not show any differential expression in PEA technology. So, how have you considered those proteins for the next phase, i.e., validation using MRM technology?

Response to referees:

We would like to thank the referee again for taking the time to review our manuscript. Below is the point-by-point response to each of those comments.

Reviewer #1

Remarks to the Author:

Author identified candidate pathogenesis-related proteins by comparing the proteome profiles measured via PEA technology between the AMS4k and AMS1k groups. In addition, they validated their initial candidates via MRM technology in an expanded cohort (AMS4k, n = 30; AMS1k, n = 30, including the 10 paired samples measured by PEA). Could you tell the **incidence** of AMS ?

We thank the reviewer for the constructive suggestion. According to our natural-observation study, 30 among 53 subjects were diagnosed as AMS based on the Louise Lake Acute Mountain Sickness Score (LLS). Therefore, the incidence rate of AMS is 57% in our study, which is among the interval (25- 94%) of reported incidence rates of AMS by other studies shown in the Introduction section.

We have clarified this in the “Study design for exploring AMS” part of Results section in our revised manuscript as follow:

“Among the 53 subjects, 30 subjects (57%) were diagnosed as AMS.”

Reviewer #2

Remarks to the Author:

The authors have done a wonderful job!

I have no more comments. Thank you for taking out time and addressing all the queries raised.

We thank the reviewer for the very kind comment describing our study as a “wonderful” job.

Reviewer #3

Remarks to the Author:

The authors has provided **well revised** version of the manuscript and my concerns were all addressed.

We thank the reviewer for positive comments on our revised manuscript.

Reviewer #4

Remarks to the Author:

Most of the comments have been addressed. I need more clarification related to my 1st comment.

1. How many samples overlapped between the discovery and validation phases? Could you show the trend of proteins level of some common proteins identified by PEA and MRM

technology?

We have clarified that a total of 20 samples (AMS, n = 10; nAMS, n = 10) overlapped between the discovery and validation phases. We have also added a heatmap of 23 common proteins identified by PEA and MRM technology in Supplementary Fig. 3.

Comment: Most of the proteins did not show any differential expression in PEA technology. So, how have you considered those proteins for the next phase, i.e., validation using MRM technology?

Comment in the attachment:

Most of the proteins did not show any differential expression in PEA technology. So, how have you considered those proteins for the next phase, i.e., validation using MRM technology? Please show the bar graph of those proteins and the no. of the sample; proteins are showing the same trend.

We thank the reviewer for the constructive suggestions on our manuscript. We identified 887 proteins by PEA technology, while only 47 proteins were significantly different (q-value < 0.05). We validated the differentially expressed proteins with MRM technology. We do not plan to validate the proteins with no significant differences identified by PEA, considering the limited budget.

We have plotted the bar graph of 23 proteins with the same trend between AMS1k (red bars) and AMS4k (green bars) groups identified by PEA and MRM technology as follows (Figure 1). The horizontal axis represents the sample ID and the vertical axis represents the normalized abundance of protein per sample. We recommended the reviewer to check the statistical results (PEA DEP and Pathogenesis sheets in Supplementary Data 3) as the barplot is unclear to visualize the results.

Figure 1. a) Bar Plot of the normalized abundance of 23 proteins identified by PEA. b) Bar plot of the normalized abundance of 23 proteins identified by MRM. These 23 proteins showed the same changing trend between AMS1k (red bars) and AMS4k (green bars) groups identified by PEA and MRM technology.